# Turing–Hopf Bifurcation Analysis in a Diffusive Ratio-Dependent Predator–Prey Model with Allee Effect and Predator Harvesting

**DOI:** 10.3390/e26010018

**Published:** 2023-12-22

**Authors:** Meiyao Chen, Yingting Xu, Jiantao Zhao, Xin Wei

**Affiliations:** Ecological Restoration and Resource Utilization for Cold Region, School of Mathematical Science, Heilongjiang University, Harbin 150080, China; meiyaochen2021@163.com (M.C.); xyingt1999@163.com (Y.X.); zhaojt@hlju.edu.cn (J.Z.)

**Keywords:** Turing–Hopf bifurcation, stability, diffusion, predator–prey model, harvesting rate

## Abstract

This paper investigates the complex dynamics of a ratio-dependent predator–prey model incorporating the Allee effect in prey and predator harvesting. To explore the joint effect of the harvesting effort and diffusion on the dynamics of the system, we perform the following analyses: (a) The stability of non-negative constant steady states; (b) The sufficient conditions for the occurrence of a Hopf bifurcation, Turing bifurcation, and Turing–Hopf bifurcation; (c) The derivation of the normal form near the Turing–Hopf singularity. Moreover, we provide numerical simulations to illustrate the theoretical results. The results demonstrate that the small change in harvesting effort and the ratio of the diffusion coefficients will destabilize the constant steady states and lead to the complex spatiotemporal behaviors, including homogeneous and inhomogeneous periodic solutions and nonconstant steady states. Moreover, the numerical simulations coincide with our theoretical results.

## 1. Introduction

The predator–prey interaction is a significant topic in the studies of populations, communities, and ecosystems and has attracted much attention from scholars. Since the introduction of the classical Lotka–Volterra model, predator–prey models have been continuously improved and developed, but there are still many ecological problems that need attention and solving [1,2,3,4]. We can use the following form of an ordinary differential equation to represent the predator–prey system
$$\begin{array}{c} \left\{\begin{array}{c} \frac{du}{dT}=f_{1}\left(u\right)+p\left(u,v\right)v,\\ \frac{dv}{dT}=f_{2}\left(v\right)+cp\left(u,v\right)v, \end{array}\right. \end{array}$$
where *u* and *v* are the densities of prey and predators, respectively. Here, $f_{1}\left(u\right)$ is the natural growth rate of the prey population in the absence of predators, $f_{2}\left(v\right)$ is the predator growth rate without prey, *c* is the conversion rate from predation, and p(u,v) is the functional response. Generally, functional responses can be divided into two categories: prey-dependent functional response and predator-dependent functional response. Prey-dependent functional response means that p(u,v) is the function of prey density only; see [4,5,6,7], for example. However, in predator-dependent predator–prey models, the function p(u,v) depends on the densities of both predator and prey. In [8], the authors proposed that a functional response should depend on the ratio of prey-to-predator density, which is supported by several laboratory observations and has been widely used in predator–prey systems. Since then, the ratio-dependent predator–prey models have gained much attention over many decades. Compared with the traditional prey-dependent predator–prey models, the ratio-dependent models can exhibit richer and more reasonable dynamical behaviors [9,10,11,12,13].

In fact, there are still other factors, as well as the functional response, that should be considered in the study of the predator–prey systems. Stephens et al. [14] suggested that the Allee effect, proposed by Allee [15], can be defined as a positive relationship between any component of individual fitness and the number or density of conspecifics. In the 2000s, Kramer et al. [16] suggested that the Allee effect can be caused by mate limitation, predator satiation, cooperative feeding or defense, habitat alternation, dispersal, etc. Their study showed that the Allee effect plays a key role in numerous systems. Later, Merdan [17] studied the effect of the Allee effect on the stability of a Lotka–Voterra model. The study demonstrated that the presence of the Allee effect makes the system take a longer time to reach the stable equilibrium and reduces the densities of both prey and predators at the stable equilibrium. Due to the significant effect on population dynamics, the Allee effect has gained increasing attention; see [18,19,20,21,22,23,24,25,26,27], for example. In view of human needs, the harvesting of biological resources should also be taken into account in predator–prey models. In [28], Xiao et al. considered the ratio-dependent predator–prey model with constant harvesting of predators as follows:(1)$$\left\{\begin{array}{c} \frac{du}{dT}=u\left(1-u\right)+\frac{auv}{u+v},\\ \frac{dv}{dT}=v\left(-d+\frac{bu}{u+v}\right)-h, \end{array}\right.$$
where *h* represents the harvesting rate; for more background about (Equation 1), we refer readers to [28] and the references therein. For system (Equation 1), the authors obtained the occurrence of numerous types of bifurcations, including the Bogdanov–Takens bifurcation, the saddle-node bifurcation, and the supercritical and subcritical Hopf bifurcations. From the perspective of biology, it is more reasonable that the harvesting rate function should be proportional to the harvested population. In [29], Chakraborty et al. proposed the modified ratio-dependent predator–prey system with nonconstant predator harvesting as follows:(2)$$\left\{\begin{array}{c} \frac{du}{dT}=ru\left(1-\frac{u}{K}\right)+\frac{\alpha uv}{u+av},\\ \frac{dv}{dT}=\frac{\alpha b_{0}u}{u+av}-dv-hv, \end{array}\right.$$
where h>0 is the harvesting rate; for more details concerning system (Equation 2), we refer readers to [29] and the references therein. This study revealed that when the harvesting rate is very high or low, the predator will eventually be extinct. In [30], the authors discussed the local stability of system (Equation 2). They gave the conditions under which the interior equilibrium is stable or unstable, a focus or a center. Their study showed that the predator harvesting rate plays a key role in the stability of the interior equilibrium of system (Equation 2), and the presence of predator harvesting makes system (Equation 2) exhibit much richer dynamical behaviors.

As is well known, the predators and prey distribute inhomogeneously in different locations; therefore, diffusion should be taken into account in ecological and biological models. Based on the fact that diffusion may destabilize the steady state and induce the occurrence of Turing instability, many scholars have investigated the diffusive predator–prey systems; see [13,31,32,33,34], for example. Although the ratio-dependent predator–prey system has been extensively investigated, a study concerning the system incorporating the Allee effect, predator harvesting, and diffusion has not been seen yet. Based on this, we consider a diffusive system with the Allee effect in prey and predator harvesting as follows:(3)$$\begin{array}{c} \left\{\begin{array}{c} \frac{\partial u}{\partial T}-d_{1}\Delta u=\frac{4\gamma }{{\left(K-Q_{0}\right)}^{2}}u\left(u-Q_{0}\right)\left(K-u\right)-G\frac{uv}{u+v},\\ \frac{\partial v}{\partial T}-d_{2}\Delta v=\mu G\frac{uv}{u+v}-Lv-Hv, \end{array}\right. \end{array}$$
where *K* is the carrying capacity for the prey, γ is the maximum per capita growth rate of the prey, *G* is the capturing rate, μ is the conversion rate, *L* is the predator death rate, and *H* is the harvesting rate. Further, $Q_{0}\left(Q_{0}<K\right)$ is the Allee threshold, and $d_{1}$ and $d_{2}$ are the diffusion coefficients. Let
$$\begin{array}{c} \hat{u}=\frac{u}{K},\hat{v}=\frac{v}{K},t=\frac{G}{T},a=\frac{4\gamma K^{2}}{G{\left(K-Q_{0}\right)}^{2}},\\ b=\frac{Q_{0}}{K},c=\mu ,d=\frac{L}{G},h=\frac{H}{G},\;D_{0}=\frac{d_{1}}{d_{2}}, \end{array}$$
and, by dropping the hats of the notations, we can obtain the corresponding diffusive system with a homogeneous Neumann boundary and initial conditions as follows:(4)$$\begin{array}{c} \left\{\begin{array}{cc} \frac{\partial u}{\partial t}-D_{0}\Delta u=au\left(u-b\right)\left(1-u\right)-\frac{uv}{u+v}, & x\in (0,l\pi ),\\ \frac{\partial v}{\partial t}-\Delta v=\frac{cuv}{u+v}-dv-hv, & x\in (0,l\pi ),\\ \frac{\partial u}{\partial \nu }=0,\;\frac{\partial v}{\partial \nu }=0, & x=0,l\pi ,\\ u\left(x,0\right)=u_{0}\left(x\right)\geq 0,v\left(x,0\right)=v_{0}\left(x\right)\geq 0, & x\in (0,l\pi ). \end{array}\right. \end{array}$$ For system (Equation 4) without the Allee effect, Gao et al. [34] studied the existence and properties of a Hopf bifurcation, provided the conditions for the occurrence of Turing instability induced by diffusion, and proved the existence and non-existence of the non-constant steady states. When the harvesting term in (Equation 4) is absent, i.e., h=0, Rao and Kang [13] investigated the effect of diffusion and the Allee effect on the dynamical complexity of the system. Their results reveal that the strength of the Allee effects plays a key role in the formation of distinct spatial patterns.

In this paper, we aim to explore the joint effect of diffusion and harvesting effort on the dynamics in system (Equation 4). Notice the fact that the term $\frac{uv}{u+v}$ has no definition at (0,0); we assume that $\frac{uv}{u+v}{|}_{\left(0,0\right)}=0$ as in [9,10]. The rest of this article is organized as follows. In Section 2, we first discuss the existence of positive equilibria. Then, we investigate the dynamics of the ODE system corresponding to system (Equation 4). In particular, using the harvesting rate as the bifurcation parameter, we study the stability of the positive equilibria, verify the existence of a Hopf bifurcation, and derive the explicit formulas for determining the properties of the bifurcating periodic solutions by applying the center manifold theory and normal form method. In Section 3, we give the sufficient conditions for the occurrence of the Turing–Hopf bifurcation. In Section 4, to illustrate the complex dynamics of system (Equation 4), we calculate the normal form near the Turing–Hopf bifurcation point. In Section 5, we give some numerical simulations to illustrate our theoretical results.

## 2. Dynamics of the ODE Model

When diffusion is absent, system (Equation 4) becomes
(5)$$\begin{array}{c} \left\{\begin{array}{c} \frac{du}{dt}=au\left(1-u\right)\left(u-b\right)-\frac{uv}{u+v},\\ \frac{dv}{dt}=\frac{cuv}{u+v}-dv-hv. \end{array}\right. \end{array}$$ It is easy to see that system (Equation 5) always has three boundary equilibria: $E_{0}=\left(0,0\right),E_{1}=\left(b,0\right),\mathrm{and}\;E_{2}=\left(1,0\right)$. Obviously, the interior equilibria should satisfy the following equation
(6)$$\begin{array}{c} \left\{\begin{array}{c} a\left(1-u\right)\left(u-b\right)=\frac{v}{u+v},\\ \frac{cu}{u+v}=d+h, \end{array}\right. \end{array}$$
which yields to
$$u^{2}-\left(1+b\right)u+b+\frac{c-d-h}{ac}=0.$$ Therefore, when the following condition
$$\left(H1\right):d+h<c\leq 1,\;ac{\left(b-1\right)}^{2}>4\left(c-d-h\right)$$
is satisfied, system (5) has two interior equilibria $E_{1}^{*}\left(u_{1}^{*},v_{1}^{*}\right)$ and $E_{2}^{*}\left(u_{2}^{*},v_{2}^{*}\right)$, where $v_{i}^{*}=\frac{c-d-h}{d+h}u_{i}^{*}\;\left(i=1,2\right)$, and
(7)$$\begin{array}{cc} u_{1}^{*}= & \frac{b+1-\sqrt{{\left(b-1\right)}^{2}-\frac{4(c-d-h)}{ac}}}{2},\\ u_{2}^{*}= & \frac{b+1+\sqrt{{\left(b-1\right)}^{2}-\frac{4(c-d-h)}{ac}}}{2}. \end{array}$$

Then, we analyze the stability of all equilibria of system (Equation 5). Note that the Jacobian matrix cannot be evaluated at $E_{0}$ since $\frac{uv}{u+v}$ is not differentiable at (0,0). We can obtain, from the first equation of (Equation 5), that
$$u\left(t\right)=u\left(0\right)e^{\int_{0}^{t}\left(\left(u\left(t\right)-b\right)\left(1-u\left(t\right)\right)-\frac{v(s)}{u(s)+v(s)}\right)ds}.$$ Similarly, from the second equation of system (Equation 5), we have that
$$v\left(t\right)=v\left(0\right)e^{\int_{0}^{t}\left(\frac{cu(s)}{u(s)+v(s)}-d-h\right)ds},$$
which means u(t)>0 and v(t)>0 when u(0)>0, v(0)>0. From the equation for the prey density, we have for $0<u<u_{0}$,
$$\frac{du}{dt}<-au\left(u-1\right)\left(u-b\right)<-au\left(u_{0}-1\right)\left(u_{0}-b\right)=-auA$$
where $u\left(0\right)=u_{0}<b$, $A=\left(u_{0}-1\right)\left(u_{0}-b\right)>0$. It can be proven that $u\left(t\right)<u_{0}e^{-aAt}$, thus $\lim_{t\to +\infty }u\left(t\right)=0$. Notice that
$$\frac{dv}{dt}=\left(\frac{c}{1+\frac{v}{u}}-d-h\right)v,$$
so we mainly consider two cases:

Case I: If v(t) has infinite extremum for t>0, then the maximal values are determined by dv/dt=0, which can be given by
$$v_{\max}\left(t_{k}\right)=\frac{c-d-h}{d+h}u_{\max}\left(t_{k}\right).$$ Since $\lim_{t\to +\infty }u\left(t\right)=0$, we can obtain that $\lim_{t\to +\infty }v\left(t\right)=0$.

Case II: If v(t) has finite extremum when t>0, then there exists a $T_{0}>0$ such that v(t) is a monotonic function of *t* when $t>T_{0}$. Notice that $\lim_{t\to +\infty }\frac{v}{u}=\infty$, we have $\lim_{t\to +\infty }\frac{dv}{dt}<0$. Therefore, v(t) is monotonically decreasing for $t>T_{0}$, and we claim that $\lim_{t\to +\infty }v\left(t\right)=0$. Otherwise, $\lim_{t\to +\infty }v\left(t\right)=B>0$. Then, for any ε>0, we have
$$\frac{dv}{dt}<v\left(\frac{c}{1+\frac{B}{u}}-d-h\right)<-v\left(1-\varepsilon \right)\left(d+h\right)$$
for a sufficiently large *t*. It is easy to prove that
$$v\left(t\right)<v_{0}\;e^{-(1-\varepsilon )(d+h)t},$$
which implies that v(t)→0 when t→+∞. It contradicts the assumption B>0. Consequently, $\lim_{t\to +\infty }v\left(t\right)=0$. Based on the above discussion, we obtain that $E_{0}$ is locally asymptotically stable. For $E_{i}\;\left(i=1,2\right)$, the corresponding Jacobian matrix can be given by
$$J{|}_{E_{1}}=\left(\begin{array}{cc} ab(1-b) & -1\\ 0 & c-(d+h) \end{array}\right),$$
$$J{|}_{E_{2}}=\left(\begin{array}{cc} -a(1-b) & -1\\ 0 & c-(d+h) \end{array}\right),$$
respectively, so we can obtain the local stability of boundary equilibria as follows.

**Lemma 1.** *For system *(Equation 5)*:*(*i*) *$E_{0}$ is a stable node;*(*ii*) *If c>d+h, then $E_{1}$ is a source, and $E_{2}$ is a saddle;*(*iii*) *If c<d+h, then $E_{1}$ is a saddle, and $E_{2}$ is a stable node.*

Next, we analyze the local stability of $E_{i}^{*}\left(i=1,2\right)$. The Jacobian Matrices around $E_{i}^{*}\;\left(i=1,2\right)$ can be written as
(8)$$J{|}_{E_{i}^{*}}=\left(\begin{array}{cc} au_{i}^{*}\left(1+b-2u_{i}^{*}\right)+\frac{u_{i}^{*}v_{i}^{*}}{{\left(u_{i}^{*}+v_{i}^{*}\right)}^{2}}\; & -\frac{u_{i}^{*2}}{{\left(u_{i}^{*}+v_{i}^{*}\right)}^{2}}\\ \frac{cv_{i}^{*2}}{{\left(u_{i}^{*}+v_{i}^{*}\right)}^{2}}\; & -\frac{cu_{i}^{*}v_{i}^{*}}{{\left(u_{i}^{*}+v_{i}^{*}\right)}^{2}} \end{array}\right),\;i=1,2,$$
and
(9)$$\mathrm{Tr}J{|}_{E_{i}^{*}}=au_{i}^{*}\left(1+b-2u_{i}^{*}\right)+\frac{u_{i}^{*}v_{i}^{*}\left(1-c\right)}{{\left(u_{i}^{*}+v_{i}^{*}\right)}^{2}},\;\;i=1,2,$$
(10)$$\mathrm{Det}J{|}_{E_{i}^{*}}=\frac{acu_{i}^{*2}v_{i}^{*}\left[2u_{i}^{*}-\left(1+b\right)\right]}{{\left(u_{i}^{*}+v_{i}^{*}\right)}^{2}},\;\;i=1,2.$$ According to Equations (Equation 7) and (Equation 10), we can conclude that
$$\mathrm{Tr}J{|}_{E_{1}^{*}}>0,\;\;\mathrm{Det}J{|}_{E_{1}^{*}}<0,\;\;\mathrm{Det}J{|}_{E_{2}^{*}}>0,$$
which implies that $E_{1}^{*}$ is always a saddle, and the stability of $E_{2}^{*}$ is determined by the sign of $\mathrm{Tr}J{|}_{E_{2}^{*}}$. Substituting the expressions for $u_{2}^{*}$ and $v_{2}^{*}$, we have
(11)$$\mathrm{Tr}J{|}_{E_{2}^{*}}=-\frac{a\left(b+1+\sqrt{{\left(b-1\right)}^{2}-\frac{4(c-d-h)}{ac}}\right)}{2}\left(\sqrt{{\left(b-1\right)}^{2}-\frac{4(c-d-h)}{ac}}\right)+\frac{\left(c-d-h)(d+h)(1-c\right)}{c^{2}}.$$ When (H1) holds, we can obtain that
$$\begin{array}{cc} \; & \lim_{h\to {\left(c-d\right)}^{-}}\mathrm{Tr}J{|}_{E_{2}^{*}}=-a\left(1-b\right)<0,\\ \; & \frac{{d\mathrm{Tr}J|}_{E_{2}^{*}}}{dh}=-\frac{b+1}{c\sqrt{{\left(b-1\right)}^{2}-\frac{4(c-d-h)}{ac}}}-\frac{2(1-c)(d+h)+c}{c^{2}}-1<0. \end{array}$$ Note that $ac{\left(b-1\right)}^{2}>4\left(c-d-h\right)$, we obtain that
$$\lim_{h\to {\left(\frac{4\left(c-d\right)-ac{\left(b-1\right)}^{2}}{4}\right)}^{+}}\mathrm{Tr}J{|}_{E_{2}^{*}}=\frac{a{\left(b-1\right)}^{2}\left[4c-ac{\left(b-1\right)}^{2}\right]\left(1-c\right)}{16c}>0$$
provided $4\left(c-d\right)-ac{\left(b-1\right)}^{2}>0$. So, under the assumptions (H1) and $4\left(c-d\right)-ac{\left(b-1\right)}^{2}>0$, (Equation 11) has a unique positive zero $h_{H}\in \left(\frac{4\left(c-d\right)-ac{\left(b-1\right)}^{2}}{4},c-d\right)$ such that $\mathrm{Tr}J{|}_{E_{2}^{*}}>0$ when $h<h_{H}$, while $\mathrm{Tr}J{|}_{E_{2}^{*}}<0$ when $h>h_{H}$. Denote
$$\left(H2\right):4\left(c-d\right)-ac{\left(b-1\right)}^{2}>0.$$ Summarizing the previous discussion, we conclude that:

**Theorem 1.** *Assume that (H1) is satisfied. Then, system *(Equation 5)* has two positive equilibria $E_{i}^{*}\;\left(i=1,2\right)$. Furthermore:*(*i*) *$E_{1}^{*}$ is unstable;*(*ii*) *If (H2) holds, then there exists a unique $h_{H}\in \left(\frac{4\left(c-d\right)-ac{\left(b-1\right)}^{2}}{4},c-d\right)$ such that $E_{2}^{*}$ is asymptotically stable when $h>h_{H}$, while $E_{2}^{*}$ is unstable when $h<h_{H}$, where $h_{H}$ is the positive zero of *(Equation 11)*.*

In fact, we can also obtain the conclusion as follows.

**Lemma 2.** *Assume that (H1) and (H2) hold. Then, system *(Equation 5)* undergoes a Hopf bifurcation at $E_{2}^{*}$ as $h=h_{H}$.*

**Proof.** Denote
(12)$${J|}_{E_{2}^{*}}=\left(\begin{array}{cc} a_{11}\; & a_{12}\\ a_{21}\; & a_{22} \end{array}\right),$$
where
(13)$$\begin{array}{cc} a_{11} & =au_{i}^{*}\left(1+b-2u_{i}^{*}\right)+\frac{u_{i}^{*}v_{i}^{*}}{{\left(u_{i}^{*}+v_{i}^{*}\right)}^{2}},\\ a_{12} & =-\frac{u_{i}^{*2}}{{\left(u_{i}^{*}+v_{i}^{*}\right)}^{2}},\;a_{21}=\frac{cv_{i}^{*2}}{{\left(u_{i}^{*}+v_{i}^{*}\right)}^{2}},\;a_{22}=-\frac{cu_{i}^{*}v_{i}^{*}}{{\left(u_{i}^{*}+v_{i}^{*}\right)}^{2}}. \end{array}$$ Let λ(h)=κ(h)±iω(h) be a pair of complex roots of
$$\lambda^{2}-\left(a_{11}+a_{22}\right)\lambda +\left(a_{11}a_{22}-a_{12}a_{21}\right)=0,$$
satisfying $\kappa (h_{H}|)=0$. Then,
$$\kappa \left(h\right)=\frac{a_{11}+a_{22}}{2},\omega \left(h\right)=\frac{1}{2}\sqrt{-4a_{12}a_{21}-{\left(a_{11}-a_{22}\right)}^{2}}.$$ Since
$$\kappa^{\prime }\left(h_{H}\right)=\frac{1}{2}\frac{{d\mathrm{Tr}J|}_{E_{2}^{*}}}{dh}<0,$$
we complete the proof. □

Let $\tilde{u}=u-u_{2}^{*}$, $\tilde{v}=v-v_{2}^{*}$. For the sake of convenience, we still denote $\tilde{u}$ and $\tilde{v}$ by *u* and *v*. Thus, system (5) can be rewritten as
(14)$$\begin{array}{c} \left\{\begin{array}{c} \frac{du}{dt}=a\left(u+u_{2}^{*}\right)\left(1-\left(u+u_{2}^{*}\right)\right)\left(\left(u+u_{2}^{*}\right)-b\right)-\frac{\left(u+u_{2}^{*}\right)\left(v+v_{2}^{*}\right)}{\left(u+u_{2}^{*}\right)+\left(v+v_{2}^{*}\right)},\\ \frac{dv}{dt}=\frac{c\left(u+u_{2}^{*}\right)\left(v+v_{2}^{*}\right)}{\left(u+u_{2}^{*}\right)+\left(v+v_{2}^{*}\right)}-d\left(v+v_{2}^{*}\right)-h\left(v+v_{2}^{*}\right). \end{array}\right. \end{array}$$ Rewrite system (14) as
(15)$$\begin{array}{c} \left(\begin{array}{c} \frac{du}{dt}\\ \frac{dv}{dt} \end{array}\right){=J|}_{E_{2}^{*}}\left(\begin{array}{c} u\\ v \end{array}\right)+\left(\begin{array}{c} f(u,v,h)\\ g(u,v,h) \end{array}\right), \end{array}$$
where
$$\begin{array}{c} f\left(u,v,h\right)=a_{1}u^{2}+a_{2}uv+a_{3}v^{2}+a_{4}u^{3}+a_{5}u^{2}v+a_{6}uv^{2}+a_{7}v^{3}+\ldots ,\\ g\left(u,v,h\right)=b_{1}u^{2}+b_{2}uv+b_{3}v^{2}+b_{4}u^{3}+b_{5}u^{2}v+b_{6}uv^{2}+b_{7}v^{3}+\ldots , \end{array}$$
with
$$\begin{array}{cccccc} & a_{1}=-3au_{2}^{*}+a+ab+\frac{v_{2}^{*2}}{{\left(u_{2}^{*}+v_{2}^{*}\right)}^{3}},\; & & a_{2}=-\frac{2u_{2}^{*}v_{2}^{*}}{{\left(u_{2}^{*}+v_{2}^{*}\right)}^{3}},\; & & a_{3}=\frac{u_{2}^{*2}}{{\left(u_{2}^{*}+v_{2}^{*}\right)}^{3}},\\ & a_{4}=-a-\frac{v_{2}^{*2}}{{\left(u_{2}^{*}+v_{2}^{*}\right)}^{4}},\; & & a_{5}=\frac{2u_{2}^{*}v_{2}^{*}-v_{2}^{*2}}{{\left(u_{2}^{*}+v_{2}^{*}\right)}^{4}},\; & & a_{6}=\frac{2u_{2}^{*}v_{2}^{*}-u_{2}^{*2}}{{\left(u_{2}^{*}+v_{2}^{*}\right)}^{4}},\\ & a_{7}=-\frac{u_{2}^{*2}}{{\left(u_{2}^{*}+v_{2}^{*}\right)}^{4}},\; & & b_{1}=-\frac{cv_{2}^{*2}}{{\left(u_{2}^{*}+v_{2}^{*}\right)}^{3}},\; & & b_{2}=\frac{2cu_{2}^{*}v_{2}^{*}}{{\left(u_{2}^{*}+v_{2}^{*}\right)}^{3}},\\ & b_{3}=-\frac{cu_{2}^{*2}}{{\left(u_{2}^{*}+v_{2}^{*}\right)}^{3}}, & & b_{4}=\frac{cv_{2}^{*2}}{{\left(u_{2}^{*}+v_{2}^{*}\right)}^{4}},\; & & b_{5}=\frac{cv_{2}^{*2}-2cu_{2}^{*}v_{2}^{*}}{{\left(u_{2}^{*}+v_{2}^{*}\right)}^{4}},\\ & b_{6}=\frac{cu_{2}^{*2}-2cu_{2}^{*}v_{2}^{*}}{{\left(u_{2}^{*}+v_{2}^{*}\right)}^{4}}, & & b_{7}=\frac{cu_{2}^{*2}}{{\left(u_{2}^{*}+v_{2}^{*}\right)}^{4}}. \end{array}$$ Let
$$P:=\left(\begin{array}{cc} N\; & 1\\ M\; & 0 \end{array}\right),$$
where
$$M=-\frac{a_{21}}{\omega (h)},\;N=\frac{a_{22}a_{11}}{2\omega (h)}.$$ Then, we obtain that
$$P^{-1}{J|}_{E_{2}^{*}}P=\phi \left(h\right):=\left(\begin{array}{cc} \kappa (h)\; & -\omega (h)\\ \omega (h)\; & \kappa (h) \end{array}\right).$$ Denote
$$M_{0}:=M{{|}_{h=h_{H}},\;N_{0}:=N|}_{h=h_{H}},\;\beta_{0}:=\omega \left(h_{H}\right).$$ Using ${\left(u,v\right)}^{T}=P{\left(p,q\right)}^{T}$, system (Equation 15) becomes
(16)$$\begin{array}{c} \left(\begin{array}{c} \frac{dp}{dt}\\ \frac{dq}{dt} \end{array}\right)=\phi \left(h\right)\left(\begin{array}{c} p\\ q \end{array}\right)+\left(\begin{array}{c} f^{1}\left(p,q,h\right)\\ g^{1}\left(p,q,h\right) \end{array}\right), \end{array}$$
where
$$\begin{array}{cc} f^{1}\left(p,q,h\right)= & \frac{1}{M}g\left(Np+q,Mp,h\right)\\ = & \left(\frac{N^{3}}{M}b_{4}+N^{2}b_{5}+NMb_{6}+M^{2}b_{7}\right)p^{3}+\left(\frac{3N^{2}}{M}b_{4}+2Nb_{5}+Mb_{6}\right)p^{2}q\\ \; & +\left(\frac{N^{2}}{M}b_{1}+Nb_{2}+Mb_{3}\right)p^{2}+\left(\frac{3N}{M}b_{4}+b_{5}\right)pq^{2}+\left(\frac{2N}{M}b_{1}+b_{2}\right)pq\\ \; & +\frac{b_{4}}{M}q^{3}+\frac{b_{1}}{M}q^{2}+\cdots , \end{array}$$
$$\begin{array}{cc} g^{1}\left(p,q,h\right)= & f\left(Np+q,Mp,h\right)-\frac{N}{M}g\left(Np+q,Mp,h\right)\\ = & \left(N^{3}a_{4}+M^{3}a_{7}+N^{2}Ma_{5}+NM^{2}a_{6}-\frac{N^{4}}{M}b_{4}-N^{3}b_{5}-MN^{2}b_{6}+NM^{2}b_{7}\right)p^{3}\\ \; & +\left(3N^{2}a_{4}+2NMa_{5}+M^{2}a_{6}-\frac{N^{4}}{M}b_{4}-2N^{2}b_{5}-NMb_{6}\right)p^{2}q\\ \; & +\left(N^{2}a_{1}+NMa_{2}+M^{2}a_{3}-\frac{3N^{3}}{M}b^{4}-2N^{2}b_{5}-NMb_{3}\right)p^{2}\\ \; & +\left(3Na_{4}+Ma_{5}-\frac{3N^{2}}{M}b_{4}-Nb_{5}\right)pq^{2}+\left(2Na_{1}+Ma_{2}-\frac{2N^{2}}{M}b_{1}-Nb_{2}\right)pq\\ \; & +\left(a_{4}-\frac{N}{M}b_{4}\right)q^{3}+\left(a_{1}-\frac{N}{M}b_{1}\right)q^{2}+\cdots . \end{array}$$ The polar coordinate form of (Equation 16) can be written as:(17)$$\begin{array}{c} \left\{\begin{array}{c} \dot{\tau }=\kappa \left(h\right)\tau +\alpha \left(h\right)\tau^{3}+\cdots ,\\ \dot{\theta }=\omega \left(h\right)+\varepsilon \left(h\right)\tau^{2}+\cdots . \end{array}\right. \end{array}$$ By Taylor expanding (Equation 17) at $h=h_{H}$, we have
(18)$$\begin{array}{c} \left\{\begin{array}{c} \dot{\tau }=\kappa^{\prime }\left(h_{H}\right)\left(h-h_{H}\right)\tau +\alpha \left(h_{H}\right)\tau^{3}+o\left(\left(h-h_{H}\right)\tau ,\left(h-h_{H}\right)\tau^{3},\tau^{5}\right),\\ \dot{\theta }=\omega \left(h_{H}\right)+\omega^{\prime }\left(h_{H}\right)\left(h-h_{H}\right)+\varepsilon \left(h_{H}\right)\tau^{2}+o\left({\left(h-h_{H}\right)}^{2},\left(h-h_{H}\right)\tau^{2},\tau^{4}\right). \end{array}\right. \end{array}$$ To determine the stability of the bifurcating periodic solutions, we need to discuss the sign of $\alpha (h_{H})$, which is given by
(19)$$\begin{array}{cc} \alpha (h_{H}):= & \frac{1}{16}\left(f_{ppp}^{1}+f_{pqq}^{1}+g_{ppq}^{1}+g_{qqq}^{1}\right)\\ \; & +\frac{1}{16\beta_{0}}\left[f_{pq}^{1}\left(f_{pp}^{1}+f_{qq}^{1}\right)-g_{pq}^{1}\left(g_{pp}^{1}+g_{qq}^{1}\right)-f_{pp}^{1}g_{pp}^{1}+f_{qq}^{1}g_{qq}^{1}\right], \end{array}$$
where all partial derivatives are evaluated at the bifurcation point $\left(p,q,b\right)=\left(0,0,h_{H}\right)$, and
$$\begin{array}{cc} \; & f_{ppp}^{1}\left(0,0,h_{H}\right)=6\left(\frac{N_{0}^{3}}{M_{0}}b_{4}+N_{0}^{2}b_{5}+N_{0}M_{0}b_{6}+M_{0}^{2}b_{7}\right),f_{pqq}^{1}\left(0,0,h_{H}\right)=2\left(\frac{3N_{0}}{M_{0}}b_{4}+b_{5}\right),\\ \; & g_{ppq}^{1}\left(0,0,h_{H}\right)=2\left(3N_{0}^{2}a_{4}+2N_{0}M_{0}a_{5}+M_{0}^{2}a_{6}-\frac{3N_{0}^{3}}{M_{0}}b_{4}-2N_{0}^{2}b_{5}-N_{0}M_{0}b_{6}\right),\\ \; & g_{qqq}^{1}\left(0,0,h_{H}\right)=6\left(a_{4}-\frac{N_{0}}{M_{0}}b_{4}\right),f_{pp}^{1}\left(0,0,h_{H}\right)=2\left(\frac{N_{0}^{2}}{M_{0}}b_{1}+N_{0}b_{2}+M_{0}b_{3}\right),\\ \; & f_{pq}^{1}\left(0,0,h_{H}\right)=\frac{2N_{0}}{M_{0}}b_{1}+b_{2},f_{qq}^{1}\left(0,0,h_{H}\right)=\frac{2}{M_{0}}b_{1},\\ \; & g_{pp}^{1}\left(0,0,h_{H}\right)=2\left(N_{0}^{2}a_{1}+N_{0}M_{0}a_{2}+M_{0}^{2}a_{3}-\frac{N_{0}^{3}}{M_{0}}b_{1}-N_{0}^{2}b_{2}-N_{0}M_{0}b_{3}\right),\\ \; & g_{pq}^{1}\left(0,0,h_{H}\right)=2N_{0}a_{1}+M_{0}a_{2}-\frac{2N_{0}^{2}}{M_{0}}b_{1}-N_{0}b_{2},g_{qq}^{1}\left(0,0,h_{H}\right)=2\left(a_{1}-\frac{N_{0}}{M_{0}}b_{1}\right). \end{array}$$
Thus, we can obtain the sign of the coefficient $\alpha (h_{H})$ in (19). Note that $\kappa^{\prime }\left(h_{H}\right)<0$; we draw the following conclusions.

**Theorem 2.** *Suppose that (H1) and (H2) hold. Then, system *(5)* undergoes a Hopf bifurcation at $E_{2}^{*}$ when $h=h_{H}$. Furthermore:*(*i*) *If $\alpha (h_{H})<0$, the bifurcating periodic solutions are orbitally asymptotically stable, and periodic solutions occur as h decreases and passes $h_{H}$.*(*ii*) *If $\alpha (h_{H})>0$, the bifurcating periodic solutions are unstable, and periodic solutions occur as h decreases and passes $h_{H}$.*

## 3. Turing Instability Induced by Diffusion and Turing–Hopf Bifurcation

In this section, we discuss the effect of diffusion on the stability of $E_{2}^{*}$ and give the sufficient conditions for the occurrence of Turing instability induced by diffusion. By computation, we have that the characteristic equations corresponding to $E_{2}^{*}$ can be given by
$$\left|\begin{array}{cc} \lambda +\frac{n^{2}}{l^{2}}D_{0}-a_{11} & -a_{12}\\ -a_{21} & \lambda +\frac{n^{2}}{l^{2}}-a_{22} \end{array}\right|=0,n\in N_{0},$$
which can be equivalent to
(20)$$\Delta_{n}\left(\lambda \right)=\lambda^{2}-T_{n}\lambda +J_{n}=0,n\in N_{0}=0,1,2,\cdots ,$$
where
(21)$$\begin{array}{cc} T_{n}= & -\left(D_{0}+1\right)\frac{n^{2}}{l_{2}}+a_{11}+a_{22},\\ J_{n}= & D_{0}\frac{n^{4}}{l^{4}}-\left(a_{11}+D_{0}a_{22}\right)\frac{n^{2}}{l^{2}}+a_{11}a_{22}-a_{12}a_{21}, \end{array}$$
with $a_{ij}\;\left(i,j=1,2\right)$ defined as in (Equation 13). In the following analysis, we always assume that (H1) and (H2) are satisfied.

**Theorem 3.** 
*Assume that (H1) and (H2) hold. Then, there exists a positive integer $k_{*}$, for $n\geq k_{*}$:*
(*i*) *System *(Equation 4)* exhibits a Turing bifurcation on $D_{0}=S_{n}\left(h\right)$ when $h_{H}<h<h_{\left(n\right)}$.*(*ii*) 
*The Turing–Hopf bifurcation occurs at $E_{2}^{*}$ when $\left(h,D_{0}\right)=\left(h_{H},D_{0(n)}^{*}\right)$, where*

$$k_{*}=\left\lfloor \sqrt{\frac{l^{2}\left(c-d-h\right)\left(d+h\right)\left(1-c\right)}{c^{2}}}\right\rfloor +1,$$


$$S_{n}\left(h\right)=\frac{\left[au_{2}^{*}\left(1+b-2u_{2}^{*}\right)+\frac{\left(c-d-h)(d+h\right)}{c^{2}}\right]\frac{n^{2}}{l^{2}}+\frac{a(c-d-h)(d+h)}{c}u_{2}^{*}\left(1+b-2u_{2}^{*}\right)}{\frac{n^{4}}{l^{4}}+\frac{\left(c-d-h)(d+h\right)}{c}\frac{n^{2}}{l^{2}}},$$


$$D_{0(n)}^{*}=\frac{\frac{\left(c-d-h)(d+h\right)}{c}\left[\frac{n^{2}}{l^{2}}-\frac{\left(c-d-h)(d+h)(1-c\right)}{c^{2}}\right]}{\frac{n^{4}}{l^{4}}+\frac{\left(c-d-h)(d+h\right)}{c}\frac{n^{2}}{l^{2}}},$$


$$h_{\left(n\right)}=\left\{h>0:S_{n}\left(h\right)=0\right\},\;\;k_{0}^{*}=\left\{n:D_{0(n)}^{*}=\max_{k\in N}\left\{D_{0(k)}^{*}\right\}\right\}.$$




**Proof.** From Theorem 1, we know that Turing instability occurs as $\Delta_{n}\left(\lambda \right)$ has roots with a positive real part for some n∈N when $h>h_{H}$. From (20), $T_{n}<0$ can be satisfied provided $h>h_{H}$. Thus, we only need to seek the condition for $J_{n}=0$ to ensure the occurrence of Turing instability. In fact, $J_{n}=0$ is equivalent to
$$D_{0}=S_{n}\left(h\right):≜\frac{\left[au_{2}^{*}\left(1+b-2u_{2}^{*}\right)+\frac{\left(c-d-h)(d+h\right)}{c^{2}}\right]\frac{n^{2}}{l^{2}}+\frac{a(c-d-h)(d+h)}{c}u_{2}^{*}\left(1+b-2u_{2}^{*}\right)}{\frac{n^{4}}{l^{4}}+\frac{\left(c-d-h)(d+h\right)}{c}\frac{n^{2}}{l^{2}}}.$$ By calculation, we have
$$\begin{array}{cc} \frac{d}{dh}S_{n}\left(h\right)= & \left[\frac{n^{6}}{l^{6}}(-\frac{b+1}{c\sqrt{{\left(b-1\right)}^{2}-\frac{4(c-d-h)}{ac}}}-\frac{1}{c}-\frac{2(d+h)}{c^{2}}\right)+2\frac{n^{4}}{l^{4}}\frac{a(c-d-h)(d+h)}{c}u_{2}^{{*}^{\prime }}\left(h\right)\left(1+b-4u_{2}^{*}\right)\\ \; & +\frac{n^{2}}{l^{2}}\frac{a{\left(c-d-h\right)}^{2}{\left(d+h\right)}^{2}}{c^{2}}u_{2}^{{*}^{\prime }}\left(h\right)\left(1+b-4u_{2}^{*}\right)]\frac{1}{{\left[\frac{n^{4}}{l^{4}}+\frac{\left(c-d-h)(d+h\right)}{c}\frac{n^{2}}{l^{2}}\right]}^{2}}, \end{array}$$
where $u_{2}^{*}\left(h\right)$ is a function of *h* defined as in (Equation 7). Since $u_{2}^{*}>\left(1+b\right)/2$, $u_{2}^{{*}^{\prime }}\left(h\right)>0$, we have $\frac{d}{dh}S_{n}\left(h\right)<0$. Then, $D_{0}=S_{n}\left(h\right)$ is a monotonically decreasing function with respect to *h*. Notice that
$$\lim_{h\to h_{H}^{+}}\;S_{n}\left(h\right)=\frac{\left(d+h)(c-d-h\right)}{c}\left[\frac{n^{2}}{l^{2}}-\frac{\left(1-c)(c-d-h)(d+h\right)}{c^{2}}\right]\frac{1}{\frac{n^{4}}{l^{4}}+\frac{\left(c-d-h)(d+h\right)}{c}\frac{n^{2}}{l^{2}}}.$$ Thus, $\lim_{h\to h_{H}^{+}}\;S_{n}\left(h\right)>0$ if and only if $\frac{n^{2}}{l^{2}}-\frac{\left(1-c)(c-d-h)(d+h\right)}{c^{2}}>0$ holds true. Denote
$$D_{0(n)}^{*}=\lim_{h\to h_{H}^{+}}\;S_{n}\left(h\right),\;\;k_{*}=\left\lfloor \sqrt{\frac{l^{2}\left(c-d-h\right)\left(d+h\right)\left(1-c\right)}{c^{2}}}\right\rfloor +1,$$
we can conclude that $D_{0(n)}^{*}=\lim_{h\to h_{H}^{+}}\;S_{n}\left(h\right)>0$ for $n\geq k^{*}$. To guarantee the positiveness of $D_{0}$ on the curve $D_{0}=S_{n}\left(h\right)$, we should ensure that the condition $h<h_{\left(n\right)}$ holds. Denote the Turing bifurcation curve as $L_{n}$, that is
$$L_{n}:D_{0}=S_{n}\left(h\right),\;\mathrm{for}\;h_{H}<h<h_{\left(n\right)},\;n\geq k_{*}.$$ When $L_{n}$ intersects the critical Hopf bifurcation curve $h=h_{H}$ at $\left(h_{H},D_{0(n)}^{*}\right)$, system (3) undergoes the Turing–Hopf bifurcation at $E_{2}^{*}$ as
$$\left(h,D_{0}\right)=\left(h_{H},D_{0(n)}^{*}\right).$$ For $n\geq k_{*}$, to seek the maximum of $D_{0(n)}^{*}$, we take the derivative of $D_{0(n)}^{*}$ with respect to $n^{2}$. We can have that
$$\frac{dD_{0(n)}^{*}}{dn^{2}}=\frac{\left(c-d-h)(d+h\right)}{cl^{2}{\left[\frac{n^{4}}{l^{4}}+\frac{\left(c-d-h)(d+h\right)}{c}\frac{n^{2}}{l^{2}}\right]}^{2}}\left[-\frac{n^{4}}{l^{4}}+\frac{2n^{2}\left(1-c\right)\left(c-d-h\right)\left(d+h\right)}{c^{2}l^{2}}+\frac{\left(1-c\right){\left(c-d-h\right)}^{2}{\left(d+h\right)}^{2}}{c^{3}}\right].$$ In fact, $\frac{dD_{0(n)}^{*}}{dn^{2}}$ has the same sign as
$$\Phi \left(n^{2}\right)=-\frac{n^{4}}{l^{4}}+\frac{2n^{2}\left(1-c\right)\left(c-d-h\right)\left(d+h\right)}{c^{2}l^{2}}+\frac{\left(1-c\right){\left(c-d-h\right)}^{2}{\left(d+h\right)}^{2}}{c^{3}}$$
when (H1) holds. Let $\Phi \left(x\right)=-\frac{x^{2}}{l^{4}}+\frac{2x(1-c)(c-d-h)(d+h)}{c^{2}l^{2}}+\frac{\left(1-c\right){\left(c-d-h\right)}^{2}{\left(d+h\right)}^{2}}{c^{3}}$. We can see that
$$\lim_{x\to 0^{+}}\Phi \left(x\right)=\frac{\left(1-c\right){\left(c-d-h\right)}^{2}{\left(d+h\right)}^{2}}{c^{3}}>0,\;\lim_{x\to +\infty }\Phi \left(x\right)<0.$$ So, there must exist a $x_{*}>0$ satisfying $\Phi (x_{*})=0$ and $\Phi (x_{*})>0$ as $x\in [0,x_{*})$, while $\Phi (x_{*})<0$ as $x\in (x_{*},\infty )$. Denote $k_{m}=\left\lfloor \sqrt{x_{*}}\right\rfloor$. As $k_{m}\leq k_{*}$, $D_{0(n)}^{*}$ is a decreasing function of *n*. As $k_{m}>k_{*}$, $D_{0(n)}^{*}$ is increasing for $n\in [k_{*},k_{m}]$ and decreasing for $n\in [k_{m}+1,\infty )$. So, $D_{0(n)}^{*}$ will reach the maximum value as $k=k_{*}$, $k=k_{m}$, or $k=k_{m}+1$. Let
$$k_{0}^{*}=\left\{\begin{array}{cc} k_{*}, & \mathrm{if}\;k_{m}\leq k_{*},\\ k_{m},\; & \mathrm{if}\;k_{m}>k_{*}\;\mathrm{and}\;D_{0(k_{m})}^{*}>D_{0(k_{m+1})}^{*},\\ k_{m}+1, & \mathrm{if}\;k_{m}>k_{*}\;\mathrm{and}\;D_{0(k_{m})}^{*}<D_{0(k_{m+1})}^{*}, \end{array}\right.$$
we can obtain that $D_{0(k_{0}^{*})}^{*}=\max_{n\in N}D_{0(n)}^{*}$. Then, we complete the proof. □

## 4. Normal Forms for Turing–Hopf Bifurcation

Denote
$$\bar{u}=u-u_{2}^{*},\;\;\bar{v}=v-v_{2}^{*},$$
and drop the bars. Then, we can rewrite (4) as follows:(22)$$\begin{array}{c} \left\{\begin{array}{c} \frac{\partial u}{\partial t}=D_{0}\Delta u+a\left(u+u_{2}^{*}\right)\left(u+u_{2}^{*}-b\right)\left(1-\left(u+u_{2}^{*}\right)\right)-\frac{\left(u+u_{2}^{*}\right)\left(v+v_{2}^{*}\right)}{u+u_{2}^{*}+v+v_{2}^{*}},\\ \frac{\partial v}{\partial t}=\Delta v+\frac{c\left(u+u_{2}^{*}\right)\left(v+v_{2}^{*}\right)}{u+u_{2}^{*}+v+v_{2}^{*}}-d\left(v+v_{2}^{*}\right)-h\left(v+v_{2}^{*}\right). \end{array}\right. \end{array}$$ By setting $h=h_{H}+\mu_{1}$ and $D_{0}=D_{0}^{*}+\mu_{2}$, $\left(\mu_{1},\mu_{2}\right)$ is the Turning-Hopf singularity in the $\mu_{1}-\mu_{2}$ plane. Thus, system (4) becomes
(23)$$\begin{array}{c} \left\{\begin{array}{c} \frac{\partial u}{\partial t}=\left(D_{0}^{*}+\mu_{2}\right)\Delta u+b_{11}u+b_{12}v+b_{13}u^{2}+b_{14}uv+b_{15}v^{2}+b_{16}u^{3}+b_{17}u^{2}v+b_{18}uv^{2}+b_{19}v^{3},\\ \frac{\partial v}{\partial t}=\Delta v+b_{21}u+b_{22}v+b_{23}u^{2}+b_{24}uv+b_{25}v^{2}+b_{26}u^{3}+b_{27}u^{2}v+b_{28}uv^{2}+b_{29}v^{3}, \end{array}\right. \end{array}$$
where
$$\begin{array}{cc} b_{11}= & au_{2}^{*}\left(h_{H}\right)\left(1+b-2u_{2}^{*}\left(h_{H}\right)\right)+\frac{\left(c-d-h_{H}\right)\left(d+h_{H}\right)}{c^{2}}\\ \; & +\mu_{1}\left[\frac{a+ab-4au_{2}^{*}\left(h_{H}\right)}{\sqrt{a^{2}c^{2}{\left(b-1\right)}^{2}-4ac\left(c-d-h_{H}\right)}}+\frac{c-2d-2h_{H}}{c^{2}}\right],\\ b_{12}= & -\frac{{\left(d+h_{H}\right)}^{2}}{c^{2}}+\mu_{1}\left[\frac{-2(d+h_{H})}{c^{2}}\right],\;\\ b_{13}= & -3au_{2}^{*}\left(h_{H}+\mu_{1}\right)+a+ab+\frac{v_{+}^{*2}\left(h_{H}+\mu_{1}\right)}{{\left(u_{2}^{*}\left(h_{H}+\mu_{1}\right)+v_{2}^{*}\left(h_{H}+\mu_{1}\right)\right)}^{3}},\\ b_{14}= & -\frac{2u_{2}^{*}\left(h_{H}+\mu_{1}\right)v_{2}^{*}\left(h_{H}+\mu_{1}\right)}{{\left(u_{2}^{*}\left(h_{H}+\mu_{1}\right)+v_{2}^{*}\left(h_{H}+\mu_{1}\right)\right)}^{3}},\;\\ b_{15}= & \frac{u_{2}^{*2}\left(h_{H}+\mu_{1}\right)}{{\left(u_{2}^{*}\left(h_{H}+\mu_{1}\right)+v_{2}^{*}\left(h_{H}+\mu_{1}\right)\right)}^{3}},\\ b_{16}= & -a-\frac{v_{2}^{*}\left(h_{H}+\mu_{1}\right)}{{\left(u_{2}^{*}\left(h_{H}+\mu_{1}\right)+v_{2}^{*}\left(h_{H}+\mu_{1}\right)\right)}^{4}},\;\;\\ b_{17}= & \frac{2u_{2}^{*}\left(h_{H}+\mu_{1}\right)v_{2}^{*}\left(h_{H}+\mu_{1}\right)-v_{2}^{*2}\left(h_{H}+\mu_{1}\right)}{{\left(u_{2}^{*}\left(h_{H}+\mu_{1}\right)+v_{2}^{*}\left(h_{H}+\mu_{1}\right)\right)}^{4}},\;\;\\ b_{18}= & \frac{-u_{2}^{*2}\left(h_{H}+\mu_{1}\right)+2u_{2}^{*}\left(h_{H}+\mu_{1}\right)v_{2}^{*}\left(h_{H}+\mu_{1}\right)}{{\left(u_{2}^{*}\left(h_{H}+\mu_{1}\right)+v_{2}^{*}\left(h_{H}+\mu_{1}\right)\right)}^{4}},\;\;\\ b_{19}= & -\frac{u_{2}^{*2}\left(h_{H}+\mu_{1}\right)}{{\left(u_{2}^{*}\left(h_{H}+\mu_{1}\right)+v_{2}^{*}\left(h_{H}+\mu_{1}\right)\right)}^{4}},\\ b_{21}= & \frac{{\left(c-d-h_{H}\right)}^{2}}{c}+\mu_{1}\left[\frac{-2(c-d-h_{H})}{c}\right],\;\;\\ b_{22}= & \frac{-\left(c-d-h\right)\left(d+h_{H}\right)}{c}+\mu_{1}\left[\frac{-(c-2d-2h_{H})}{c}\right],\;\;\\ b_{23}= & -\frac{cv_{2}^{*2}\left(h_{H}+\mu_{1}\right)}{{\left(u_{2}^{*}\left(h_{H}+\mu_{1}\right)+v_{2}^{*}\left(h_{H}+\mu_{1}\right)\right)}^{3}},\;\\ b_{24}= & \frac{2cu_{2}^{*}\left(h_{H}+\mu_{1}\right)v_{2}^{*}\left(h_{H}+\mu_{1}\right)}{{\left(u_{2}^{*}\left(h_{H}+\mu_{1}\right)+v_{2}^{*}\left(h_{H}+\mu_{1}\right)\right)}^{3}},\;\\ b_{25}= & -\frac{cu_{2}^{*2}\left(h_{H}+\mu_{1}\right)}{{\left(u_{2}^{*}\left(h_{H}+\mu_{1}\right)+v_{2}^{*}\left(h_{H}+\mu_{1}\right)\right)}^{3}},\\ b_{26}= & \frac{cv_{2}^{*2}\left(h_{H}+\mu_{1}\right)}{{\left(u_{2}^{*}\left(h_{H}+\mu_{1}\right)+v_{2}^{*}\left(h_{H}+\mu_{1}\right)\right)}^{4}},\\ b_{27}= & \frac{cv_{2}^{*2}\left(h_{H}+\mu_{1}\right)-2cu_{2}^{*}\left(h_{H}+\mu_{1}\right)v_{2}^{*}\left(h_{H}+\mu_{1}\right)}{{\left(u_{2}^{*}\left(h_{H}+\mu_{1}\right)+v_{2}^{*}\left(h_{H}+\mu_{1}\right)\right)}^{4}},\;\;\\ b_{28}= & \frac{cu_{2}^{*2}\left(h_{H}+\mu_{1}\right)-2cu_{2}^{*}\left(h_{H}+\mu_{1}\right)v_{2}^{*}\left(h_{H}+\mu_{1}\right)}{{\left(u_{2}^{*}\left(h_{H}+\mu_{1}\right)+v_{2}^{*}\left(h_{H}+\mu_{1}\right)\right)}^{4}},\;\;\\ b_{29}= & c\frac{u_{2}^{*2}\left(h_{H}+\mu_{1}\right)}{{\left(u_{2}^{*}\left(h_{H}+\mu_{1}\right)+v_{2}^{*}\left(h_{H}+\mu_{1}\right)\right)}^{4}}. \end{array}$$ It follows from (20) that for (22), when $\mu_{1}=\mu_{2}=0$, $\Delta_{0}\left(\lambda \right)=0$ has a pair of purely imaginary roots $\pm i\omega_{0}$ with $\omega_{0}=\sqrt{J_{0}}$, $\Delta_{k_{0}^{*}}\left(\lambda \right)=0$ has a zero root and a negative real root $\lambda =-T_{k_{0}^{*}}$, and, if $k\neq 0,k_{0}^{*}$, all of the roots of $\Delta_{k}\left(\lambda \right)=0$ have negative real parts. For (23), we have
$$\begin{array}{c} D\left(\mu \right)=\left(\begin{array}{cc} D_{0}^{*}+\mu_{2} & 0\\ 0 & 1 \end{array}\right),\;\;L\left(\mu \right)=\left(\begin{array}{cc} b_{11} & b_{12}\\ b_{21} & b_{22} \end{array}\right), \end{array}$$
and
$$\begin{array}{c} F\left(u,v,\mu_{1},\mu_{2}\right)=\left(\begin{array}{c} b_{13}u^{2}+b_{14}uv+b_{15}v^{2}+b_{16}u^{3}+b_{17}u^{2}v+b_{18}uv^{2}+b_{19}v^{3}\\ b_{23}u^{2}+b_{24}uv+b_{25}v^{2}+b_{26}u^{3}+b_{27}u^{2}v+b_{28}uv^{2}+b_{29}v^{3} \end{array}\right). \end{array}$$ For convenience, we rewrite D(μ) and L(μ) as
(24)$$\begin{array}{c} D\left(\mu \right)=D_{0}+D_{1}^{\left(1,0\right)}\mu_{1}+D_{1}^{\left(0,1\right)}\mu_{2}, \end{array}$$
(25)$$\begin{array}{c} L\left(\mu \right)=L_{0}+L_{1}^{\left(1,0\right)}\mu_{1}+L_{1}^{\left(0,1\right)}\mu_{2}. \end{array}$$ For *L*, we have
$$\begin{array}{c} D_{0}=\left(\begin{array}{cc} D_{0}^{*} & 0\\ 0 & 1 \end{array}\right),D_{1}^{\left(1,0\right)}=\left(\begin{array}{cc} 0 & 0\\ 0 & 0 \end{array}\right),D_{1}^{\left(0,1\right)}=\left(\begin{array}{cc} 1 & 0\\ 0 & 0 \end{array}\right),\\ L_{0}=\left(\begin{array}{cc} l_{0,11} & l_{0,12}\\ l_{0,21} & l_{0,22} \end{array}\right),L_{1}^{\left(1,0\right)}=\left(\begin{array}{cc} l_{1,11} & l_{1,12}\\ l_{1,21} & l_{1,22} \end{array}\right),L_{1}^{\left(0,1\right)}=\left(\begin{array}{cc} 0 & 0\\ 0 & 0 \end{array}\right), \end{array}$$
with
$$\begin{array}{cccc} & l_{0,11}=au_{2}^{*}\left(h_{H}\right)\left(1+b-2u_{+}^{*}\left(h_{H}\right)\right)+\frac{\left(c-d-h_{H}\right)\left(d+h_{H}\right)}{c^{2}},\; & & l_{0,12}=-\frac{{\left(d+h_{H}\right)}^{2}}{c^{2}},\\ & l_{0,21}=\frac{{\left(c-d-h_{H}\right)}^{2}}{c},\; & & l_{0,22}=\frac{-\left(c-d-h\right)\left(d+h_{H}\right)}{c},\;\\ & l_{1,11}=\frac{a+ab-4au_{2}^{*}\left(h_{H}\right)}{\sqrt{a^{2}c^{2}{\left(b-1\right)}^{2}-4ac\left(c-d-h_{H}\right)}}+\frac{c-2d-2h_{H}}{c^{2}},\; & & l_{1,12}=\frac{-2(d+h_{H})}{c^{2}},\\ & l_{1,21}=\frac{-2(c-d-h_{H})}{c},\; & & l_{1,22}=\frac{-(c-2d-2h_{H})}{c}.\;\; \end{array}$$ Let
$$M_{k}\left(\lambda \right)=\left(\begin{array}{cc} \lambda +D_{0}^{*}\frac{k^{2}}{l^{2}}-l_{0,11} & -l_{0,12}\\ -l_{0,21} & \lambda +\frac{k^{2}}{l^{2}}-l_{0,22} \end{array}\right).$$ By calculation, we obtain that
$$\Phi_{1}=\left(\xi_{0},\bar{\xi_{0}}\right),\;\;\Phi_{2}=\xi_{n_{*}},$$
$$\Psi_{1}=\mathrm{col}\left(\eta_{0}^{T},\bar{\eta_{0}^{T}}\right),\;\;\Psi_{2}=\eta_{n_{*}}^{T},$$
where
$$\begin{array}{c} \xi_{0}=\left(\begin{array}{c} \xi_{01}\\ \xi_{02} \end{array}\right)=\left(\begin{array}{c} 1\\ \frac{i\omega_{0}-l_{0,11}}{l_{0,12}} \end{array}\right),\;\;\eta_{0}=\left(\begin{array}{c} \eta_{01}\\ \eta_{02} \end{array}\right)=D_{1}\left(\begin{array}{c} 1\\ \frac{i\omega_{0}-l_{0,11}}{l_{0,21}} \end{array}\right),\\ \bar{\xi_{0}}=\left(\begin{array}{c} \bar{\xi_{01}}\\ \bar{\xi_{02}} \end{array}\right),=\left(\begin{array}{c} 1\\ \frac{-i\omega_{0}-l_{0,11}}{l_{0,12}} \end{array}\right),\;\;\bar{\eta_{0}}=\left(\begin{array}{c} \bar{\eta_{01}}\\ \bar{\eta_{02}} \end{array}\right)=D_{1}\left(\begin{array}{c} 1\\ \frac{-i\omega_{0}-l_{0,11}}{l_{0,21}} \end{array}\right),\\ \xi_{n_{*}}=\left(\begin{array}{c} \xi_{n_{*}1}\\ \xi_{n_{*}2} \end{array}\right)=\left(\begin{array}{c} 1\\ \frac{D_{0}^{*}\frac{n_{*}^{2}}{l^{2}}-l_{0,11}}{l_{0,12}} \end{array}\right),\;\;\eta_{n_{*}}=\left(\begin{array}{c} \eta_{n_{*}1}\\ \eta_{n_{*}2} \end{array}\right)=D_{2}\left(\begin{array}{c} 1\\ \frac{D_{0}^{*}\frac{n_{*}^{2}}{l^{2}}-l_{0,11}}{l_{0,21}} \end{array}\right), \end{array}$$
with
$$\begin{array}{c} D_{1}=\frac{l_{0,21}l_{0,12}}{l_{0,21}l_{0,12}+{\left(i\omega_{0}-l_{0,11}\right)}^{2}},\;\;D_{2}=\frac{l_{0,21}l_{0,12}}{l_{0,21}l_{0,12}+{\left(D_{0}^{*}\frac{n_{*}^{2}}{l^{2}}-l_{0,11}\right)}^{2}}. \end{array}$$

Following the techniques in [35], by a recursive transformation, we can obtain that the normal form for the Turing–Hopf bifurcation can be given by
(26)$$\dot{z}=Bz+\left(\begin{array}{c} B_{11}\mu_{1}z_{1}+B_{21}\mu_{2}z_{1}\\ {\bar{B}}_{11}\mu_{1}z_{1}+{\bar{B}}_{21}\mu_{2}z_{1}\\ B_{13}\mu_{1}z_{3}+B_{23}\mu_{2}z_{3} \end{array}\right)+\left(\begin{array}{c} B_{210}z_{1}^{2}z_{2}+B_{102}z_{1}z_{3}^{2}\\ {\bar{B}}_{210}z_{1}^{2}z_{2}+{\bar{B}}_{102}z_{1}z_{3}^{2}\\ B_{111}z_{1}z_{2}z_{3}+B_{003}z_{3}^{3} \end{array}\right)+O(|z||\mu^{2}\left|\right),$$
where
$$\begin{array}{c} B_{11}=\eta_{0}^{T}L_{1}^{\left(1,0\right)}\xi_{0},\;B_{21}=\eta_{0}^{T}L_{1}^{\left(0,1\right)}\xi_{0},\\ B_{13}=\eta_{n_{*}}^{T}\left(-\frac{n_{*}^{2}}{l^{2}}D_{1}^{\left(1,0\right)}\xi_{n_{*}}+L_{1}^{\left(1,0\right)}\xi_{n_{*}}\right),\;B_{23}=\eta_{n_{*}}^{T}\left(-\frac{n_{*}^{2}}{l^{2}}D_{1}^{\left(0,1\right)}\xi_{n_{*}}+L_{1}^{\left(0,1\right)}\xi_{n_{*}}\right), \end{array}$$
and
$$\begin{array}{c} B_{210}=C_{210}+\frac{3}{2}\left(D_{210}+E_{210}\right),\;\;B_{102}=C_{102}+\frac{3}{2}\left(D_{102}+E_{102}\right),\\ B_{111}=C_{111}+\frac{3}{2}\left(D_{111}+E_{111}\right),\;\;B_{003}=C_{003}+\frac{3}{2}\left(D_{003}+E_{003}\right). \end{array}$$ Next, we need to calculate $C_{ijk}$, $D_{ijk}$, and $E_{ijk}$.
$$F_{j_{1}j_{2}}={\left(F_{j_{1}j_{2}}^{\left(1\right)},F_{j_{1}j_{2}}^{\left(2\right)}\right)}^{T},$$
with
$$F_{j_{1}j_{2}}^{\left(k\right)}=\frac{\partial F^{\left(k\right)}\left(0,0,0,0\right)}{\partial u^{j_{1}}\partial v^{j_{2}}},\;k=1,2,\;j_{1}+j_{2}=2,3.$$ Then, we can figure out by calculation that
A200=F20+2ξ02F11+ξ022F02=A¯020,A002=F20+2ξn∗2F11+ξn∗22F02,A110=2(F20+2Re(ξ02)F11+|ξ02|2F02),A101=2(F20+(ξ02+ξn∗2)F11+ξ02ξn∗2F02)=A¯011,A210=3(F30+(2ξ02+ξ¯02)F21+(ξ022+2|ξ02|2)F12+|ξ02|2ξ02F03),A102=3(F30+(ξ02+2ξn∗2)F21+(ξn∗22+2ξ02ξn∗2)F12+ξ02ξn∗22F03),A111=6(F30+(ξn∗2+2Reξ02)F21+(|ξ02|2+2ξn∗2Reξ02)F12+|ξ02|2ξn∗2F03),A003=F30+3(ξn∗2F21+ξn∗22F12)+ξn∗22F03. Thus, we can obtain
$$C_{210}=\frac{1}{6l\pi }\eta_{0}^{T}A_{210},\;\;C_{102}=\frac{1}{6l\pi }\eta_{0}^{T}A_{102},$$
$$C_{111}=\frac{1}{6l\pi }\eta_{n_{*}}^{T}A_{111},\;\;C_{003}=\frac{1}{6l\pi }\eta_{n_{*}}^{T}A_{003},$$
$$\begin{array}{cc} D_{210}= & \frac{1}{6l\pi i\omega_{0}}\left[-\left(\eta_{0}^{T}A_{200}\right)\left(\eta_{0}^{T}A_{110}\right)+\left(\eta_{0}^{T}A_{110}\right)\left(\bar{\eta_{0}^{T}}A_{110}\right)+\frac{2}{3}\left(\eta_{0}^{T}A_{020}\right)\left(\bar{\eta_{0}^{T}}A_{200}\right)\right],\\ D_{102}= & \frac{1}{6l\pi i\omega_{0}}\left[-2\left(\eta_{0}^{T}A_{200}\right)\left(\eta_{0}^{T}A_{002}\right)+\left(\eta_{0}^{T}A_{110}\right)\left(\bar{\eta_{0}^{T}}A_{002}\right)+2\left(\eta_{0}^{T}A_{002}\right)\left(\bar{\eta_{{k_{0}}^{*}}^{T}}A_{101}\right)\right],\\ D_{111}= & \frac{1}{6l\pi i\omega_{0}}[\left(\eta_{n_{*}}^{T}A_{011}\right)\left(\bar{\eta_{0}^{T}}A_{110}\right)-\left(\eta_{n_{*}}^{T}A_{101}\right)\left(\eta_{0}^{T}A_{110}\right),\\ D_{003}= & \frac{1}{6l\pi i\omega_{0}}[\left(\eta_{n_{*}}^{T}A_{011}\right)\left(\bar{\eta_{0}^{T}}A_{002}\right)-\left(\eta_{n_{*}}^{T}A_{101}\right)\left(\eta_{0}^{T}A_{002}\right),\\ E_{210}= & \frac{1}{6}\eta_{0}^{T}\left[S_{yz_{1}}\langle h_{00110}\rangle +S_{yz_{2}}\langle h_{00200}\rangle \right],\\ E_{102}= & \frac{1}{6}\eta_{0}^{T}\left[S_{yz_{1}}\langle h_{00002}\rangle +S_{yz_{3}}\langle h_{n_{*}0101}\rangle \right],\\ E_{111}= & \frac{1}{6}\eta_{n_{*}}^{T}\left[S_{yz_{1}}\langle h_{0n_{*}011}\rangle +S_{yz_{2}}\langle h_{0n_{*}10}\rangle +S_{yz_{3}}\langle h_{n_{*}n_{*}110}\rangle \right],\\ E_{003}= & \frac{1}{6}\eta_{n_{*}}^{T}\left[S_{yz_{3}}\langle h_{n_{*}n_{*}002}\rangle \right], \end{array}$$
with
$$\begin{array}{c} S_{yz_{1}}=\left(\begin{array}{cc} 2b_{13}+b_{14}\xi_{02} & b_{14}+2b_{15}\xi_{02}\\ 2b_{23}+b_{24}\xi_{02} & b_{24}+2b_{25}\xi_{02} \end{array}\right),\\ S_{yz_{2}}=\left(\begin{array}{cc} 2b_{13}+b_{14}{\bar{\xi }}_{02} & b_{14}+2b_{15}{\bar{\xi }}_{02}\\ 2b_{23}+b_{24}{\bar{\xi }}_{02} & b_{24}+2b_{25}{\bar{\xi }}_{02} \end{array}\right),\\ S_{yz_{3}}=\left(\begin{array}{cc} 2b_{13}+b_{14}{\bar{\xi }}_{0n_{*}} & b_{14}+2b_{15}{\bar{\xi }}_{0n_{*}}\\ 2b_{23}+b_{24}{\bar{\xi }}_{0n_{*}} & b_{24}+2b_{25}{\bar{\xi }}_{0n_{*}} \end{array}\right), \end{array}$$
and
$$\begin{array}{cc} \; & h_{00110}=\frac{1}{l\pi }\left\{-{\left[L_{0}\left(I_{d}\right)\right]}^{-1}A_{110}+\frac{1}{i\omega_{0}}\left[\eta_{0}^{T}A_{110}\xi_{0}+\frac{1}{3}\bar{\eta_{0}^{T}}A_{110}\bar{\xi_{0}}\right]\right\},\\ \; & h_{00200}=\frac{1}{l\pi }\left\{{\left(2i\omega_{0}-L_{0}\left(I_{d}\right)\right)}^{-1}A_{200}-\frac{1}{i\omega_{0}}\left[\eta_{0}^{T}A_{200}\xi_{0}+\frac{1}{3}\bar{\eta_{0}^{T}}A_{110}\bar{\xi_{0}}\right]\right\},\\ \; & h_{00002}=-\frac{1}{l\pi }{\left[L_{0}\left(I_{d}\right)\right]}^{-1}A_{002}+\frac{1}{l\pi i\omega_{0}}\left[\eta_{0}^{T}A_{002}\xi_{0}-\bar{\eta_{0}^{T}}A_{002}\bar{\xi_{0}}\right],\\ \; & h_{n_{*}0101}=\frac{1}{l\pi }{\left[i\omega_{0}+\frac{n_{*}^{2}}{l^{2}}D_{0}-L_{0}\right]}^{-1}A_{101}-\frac{1}{l\pi i\omega_{0}}\left[\eta_{n^{*}}^{T}A_{101}\xi_{n_{*}}\right],\\ \; & h_{0n_{*}011}=\frac{1}{l\pi }{\left[-i\omega_{0}+\frac{n_{*}^{2}}{l^{2}}D_{0}-L_{0}\right]}^{-1}A_{011}+\frac{1}{l\pi i\omega_{0}}\left[\eta_{n^{*}}^{T}A_{101}\xi_{n_{*}}\right],\\ \; & h_{n_{*}n_{*}002}=\frac{1}{2l\pi }{\left[\frac{{\left(2n_{*}\right)}^{2}}{l^{2}}D_{0}-L_{0}\right]}^{-1}A_{002}+h_{00002},\\ \; & h_{0n_{*}101}=h_{n_{*}0101},\;\;h_{n_{*}n_{*}110}=h_{00110}. \end{array}$$ So, the normal form truncated to the third-order terms for the Turing–Hopf bifurcation can be written as
(27)$$\left\{\begin{array}{c} \dot{\rho }=\alpha_{1}\left(\mu \right)\rho +\kappa_{11}\rho^{3}+\kappa_{12}\rho \varsigma^{2},\\ \dot{\varsigma }=\alpha_{2}\left(\mu \right)\varsigma +\kappa_{21}\rho^{2}\varsigma +\kappa_{22}\varsigma^{3}, \end{array}\right.$$
where
α1(μ)=Re(B11)μ1+Re(B21)μ2,α2(μ)=B13μ1+B23μ2,κ11=sign(Re(B210)),κ12=Re(B102)|B003|,κ21=B111|Re(B210)|,κ22=sign(B003).

## 5. Numerical Simulations

In this section, we provide some numerical simulations to show the previous analysis.

To illustrate Theorem 2, we choose a=1.82, b=0.21, c=0.5, d=0.3, and h=0.09 such that (H1) and (H2) hold. By calculation, we have $E_{2}^{*}=\left(0.6439,0.2561\right)$, $h_{H}=0.0591$, and $\alpha (h_{H})=-0.7076<0$. According to Theorem 2, we know that (Equation 5) undergoes a Hopf bifurcation at $E_{2}^{*}$ when *h* decreases and passes $h_{H}$, and the bifurcating periodic solutions are stable, as shown in Figure 1.

Choose a=1.8,b=0.2,c=0.5,d=0.3,andl=2, such that (H1) and (H2) hold. We can obtain, from Theorem 3, that the Turing–Hopf bifurcation occurs at $\left(h_{H},D_{2(k_{0}^{*})}^{*}\right)=\left(0.05774,0.17154\right)$ and the wave number is $k_{*}=1$ (see Figure 2). When $\left(h,D_{0}\right)=\left(0.05774,0.17154\right)$, $\left(u_{2}^{*},v_{2}^{*}\right)=\left(0.6439,0.2561\right)$. By calculation, the normal form truncated to the third-order term is
(28)$$\left\{\begin{array}{c} \dot{\rho }=\rho \left(-15.8798\mu_{1}-\rho^{2}-0.2403\varsigma^{2}\right),\\ \dot{\varsigma }=r\left(-38.1544\mu_{1}-0.3003\mu_{2}+11.3740\rho^{2}+\varsigma^{2}\right). \end{array}\right.$$ Recalling that ρ>0, system (Equation 28) has the equilibria
${\tilde{A}}_{0}=\left(0,\;0\right)$,${\tilde{A}}_{1}=\left(\sqrt{-15.8798\mu_{1}},\;0\right)$, for $\mu_{1}<0$,${\tilde{A}}_{2}^{\pm }=\left(0,\;\pm \sqrt{-38.1521\mu 1-0.3002\mu_{2}}\right)$, for $38.1521\mu 1+0.3002\mu_{2}<0$,${\tilde{A}}_{3}^{\pm }=\left(\sqrt{-1.7973\mu_{1}+0.0193\mu_{2}},\;\pm \sqrt{-58.5938\mu_{1}-0.0804\mu_{2}}\right)$,
for $-1.7973\mu_{1}+0.0193\mu_{2}>0,\;58.5938\mu_{1}+0.0804\mu_{2}<0$.

Denote the bifurcation curves as
$$\begin{array}{cccc} & H_{0}:\mu_{1}=0; & & T:\mu_{2}=-127.057\mu_{1};\\ & T_{1}:\mu_{2}=-728.5753\mu_{1},\mu_{1}<0; & & \;T_{2}:\mu_{2}=92.9795\mu_{1},\mu_{1}<0. \end{array}$$ Then, we obtain the bifurcation diagram in the $\mu_{1}-\mu_{2}$ plane and the corresponding phase portraits of system (Equation 28) in the ρ−ς plane, as shown in Figure 3. Clearly, the above curves divide the $\mu_{1}-\mu_{2}$ plane into six regions, denoted as $R_{i},i=1,2\cdots ,6$ (see Figure 3). The existence and stability properties of the steady states in the six regions are listed in Table 1.

Obviously, the equilibria ${\tilde{A}}_{0}$, ${\tilde{A}}_{1}$, ${\tilde{A}}_{2}^{\pm }$, and ${\tilde{A}}_{3}^{\pm }$ of system (Equation 28) correspond to the constant equilibrium, the spatially homogeneous periodic solution, the nonconstant steady state, and the spatially inhomogeneous periodic solution of system (Equation 4). Thus, the dynamical behaviors of system (Equation 4) nearby the Turing–Hopf singularity in the $h-D_{0}$ plane can be determined by the dynamical behaviors of system (Equation 28).

In $R_{1}$, there is a stable equilibrium, ${\tilde{A}}_{0}$, in (Equation 28), which means $E_{2}^{*}$ is asymptotically stable; see Figure 4.

In $R_{2}$, there are two equilibria, ${\tilde{A}}_{0}$ and ${\tilde{A}}_{1}$ in (Equation 28). Since ${\tilde{A}}_{1}$ is stable, system (Equation 4) has a stable, spatially homogeneous periodic solution; see Figure 5.

In $R_{3}$, there are four equilibria, ${\tilde{A}}_{0}$, ${\tilde{A}}_{1}$, and ${\tilde{A}}_{3}^{\pm }$ in (Equation 28). Since ${\tilde{A}}_{3}^{+}$ and ${\tilde{A}}_{3}^{-}$ are stable, there exist spatially inhomogeneous periodic solutions; see Figure 6.

In $R_{4}$, there are six equilibria, ${\tilde{A}}_{0}$, ${\tilde{A}}_{1}$, ${\tilde{A}}_{2}^{\pm }$, and ${\tilde{A}}_{3}^{\pm }$, in (Equation 28). Since ${\tilde{A}}_{3}^{+}$ and ${\tilde{A}}_{3}^{-}$ are stable, there exist spatially inhomogeneous periodic solutions; see Figure 7.

In $R_{5}$, there are four equilibria, ${\tilde{A}}_{0}$, ${\tilde{A}}_{1}$, and ${\tilde{A}}_{2}^{\pm }$, in (Equation 28). Since ${\tilde{A}}_{2}^{+}$ and ${\tilde{A}}_{2}^{-}$ are stable, there exist spatially inhomogeneous steady states; see Figure 8.

In $R_{6}$, there are three equilibria, ${\tilde{A}}_{0}$ and ${\tilde{A}}_{2}^{\pm }$, in (Equation 28). Since ${\tilde{A}}_{2}^{+}$ and ${\tilde{A}}_{2}^{-}$ are stable, there exist spatially inhomogeneous steady states; see Figure 9.

## 6. Conclusions and Discussion

In this paper, we investigate the ratio-dependent predator–prey system with the Allee effect in prey and predator harvesting. We give a detailed analysis of the joint effect of harvesting effort and diffusion on the spatiotemporal behaviors of system (Equation 4), and our results reveal that the presence of a harvesting term makes the system exhibit more interesting dynamical behaviors.

A ratio-dependent predator–prey system with a harvesting term is a relatively new issue that has been investigated by several researchers and has yielded many interesting results. Recently, Gao et al. [34] analyzed the existence of a Hopf bifurcation induced by harvesting rate and a Turing bifurcation induced by diffusion, respectively, for systems (Equation 4) without the Allee effect. However, they did not consider the dynamics of multi-parameter synergism, which is also a major difficulty in our research.

The main contribution of our paper is a detailed analysis of bifurcation near the positive constant steady state of system (Equation 4) in the one-dimensional spatial domain (0,lπ). For the spatially homogeneous model, using the harvesting rate *h* as the bifurcation parameter, we analyze the stability of interior equilibria. By applying the center manifold theory and normal form method, we derive the formula determining the direction of the Hopf bifurcation and the stability of the bifurcating periodic solutions. For the reaction–diffusion model, we firstly verify the existence of Turing instability induced by diffusion, which reveals the existence of spatially inhomogeneous patterns, including the spatially inhomogeneous periodic solutions and non-constant steady-state solutions. Then, the normal form near the Turing–Hopf bifurcation point is derived. Our study demonstrates that as the harvesting rate *h* decreases and passes the critical value $h_{H}$, the coexistence equilibrium $E_{2}^{*}$ will lose its stability, and the Hopf bifurcation occurs. Finally, numerical simulations, which are consistent with the theoretical results, are performed to illustrate the theoretical analysis.

In ecosystems, humans, as higher animals, have the ability to harvest biological resources. Our research shows that if humans overharvest biological resources, it will lead to the unbalance of the ecosystem. Our study provides the critical harvesting rate $h_{H}$ without destroying the ecosystem, which not only ensures the health of the ecosystem but also maximizes the biological resources available to humans. At the same time, our results also show that humans can obtain better harvest times and easier harvest locations by controlling the ratio of the diffusion coefficients of prey and predators and the harvesting rate near the Turing–Hopf singularity $\left(h_{H},D_{0(n)}^{*}\right)$. This can greatly reduce the difficulty for humans in obtaining biological resources.

In [36], the authors proposed the non-continuous harvesting function as follows
$$H\left(x\right)=\left\{\begin{array}{cc} 0,\; & x<T,\\ qx,\; & x>T. \end{array}\right.$$ They assumed that the harvesting may start as the population reaches some critical value *T*. To further the study, we will consider a non-continuous harvesting function in our system, compare the theoretical results to the results in the paper, and reveal the effect of the harvesting rate on the predator population. Moreover, we may consider the effect of spatial heterogeneity on the dynamics of (Equation 4).

## Figures and Tables

**Figure 1 entropy-26-00018-f001:**
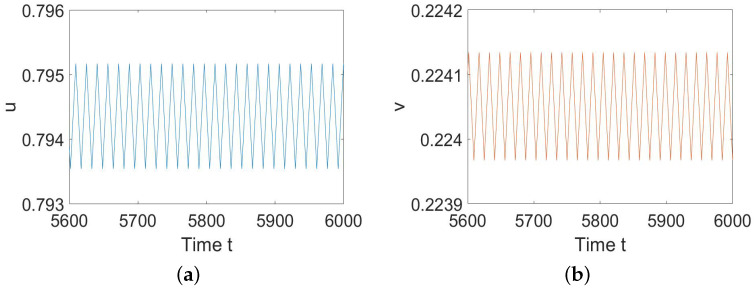
System (Equation 5) has stable periodic solutions. (**a**) represents the prey *u*, and (**b**) represents the predator *v*.

**Figure 2 entropy-26-00018-f002:**
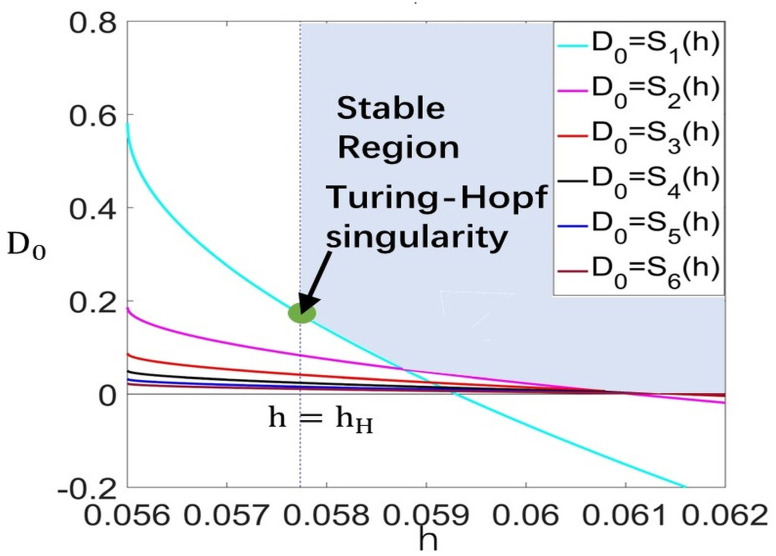
Stable region for $E_{*}$ in $h-D_{0}$ plane on region [0.056,0.07]×[−0.2,0.8] as $\left(h,D_{0}\right)=\left(0.05774,0.17154\right)$, $k^{*}=1$.

**Figure 3 entropy-26-00018-f003:**
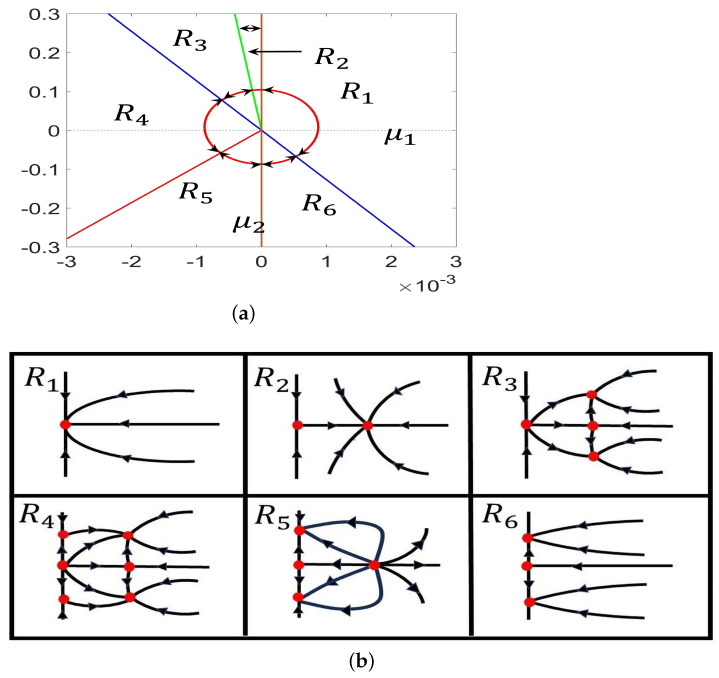
(**a**) Detailed parameter regions near the Turing–Hopf bifurcation point (0.05774,0.17154); (**b**) phase portraits in region $R_{1}-R_{6}$.

**Figure 4 entropy-26-00018-f004:**
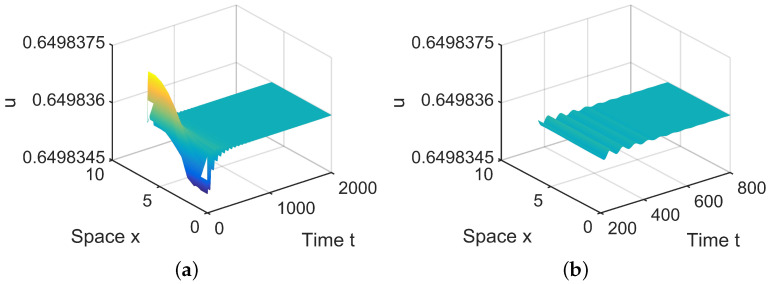
When $\left(\mu_{1},\mu_{2}\right)$ = (0.0005, 0.001) lies in $R_{1}$, $E_{2}^{*}=\left(0.6498,0.2572\right)$ is locally asymptotically stable. (**a**,**b**) represent the prey *u*, and (**c**,**d**) represent the predator *v*. The initial values are chosen as $u_{0}\left(x,t\right)=0.6498-0.000001\sin x$, $v_{0}\left(x,t\right)=0.2572+0.000001\cos x$. (**b**,**d**) are middle-term behaviors for u and v, respectively.

**Figure 5 entropy-26-00018-f005:**
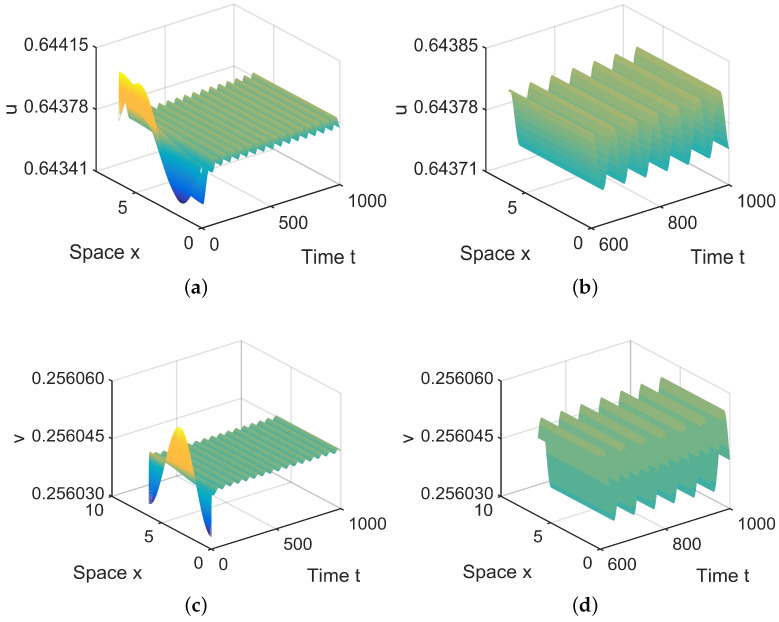
When $\left(\mu_{1},\mu_{2}\right)$ = (−0.00001, 0.1) lies in $R_{2}$, system (Equation 4) has stable, spatially homogeneous period solutions. (**a**,**b**) represent the prey *u*, and (**c**,**d**) represent the predator *v*. The initial values are chosen as $u_{0}\left(x,t\right)=0.6437-0.0002\;\sin x$, $v_{0}\left(x,t\right)=0.256-0.00025\cos x$. (**b**,**d**) are long-term behaviors for u and v, respectively.

**Figure 6 entropy-26-00018-f006:**
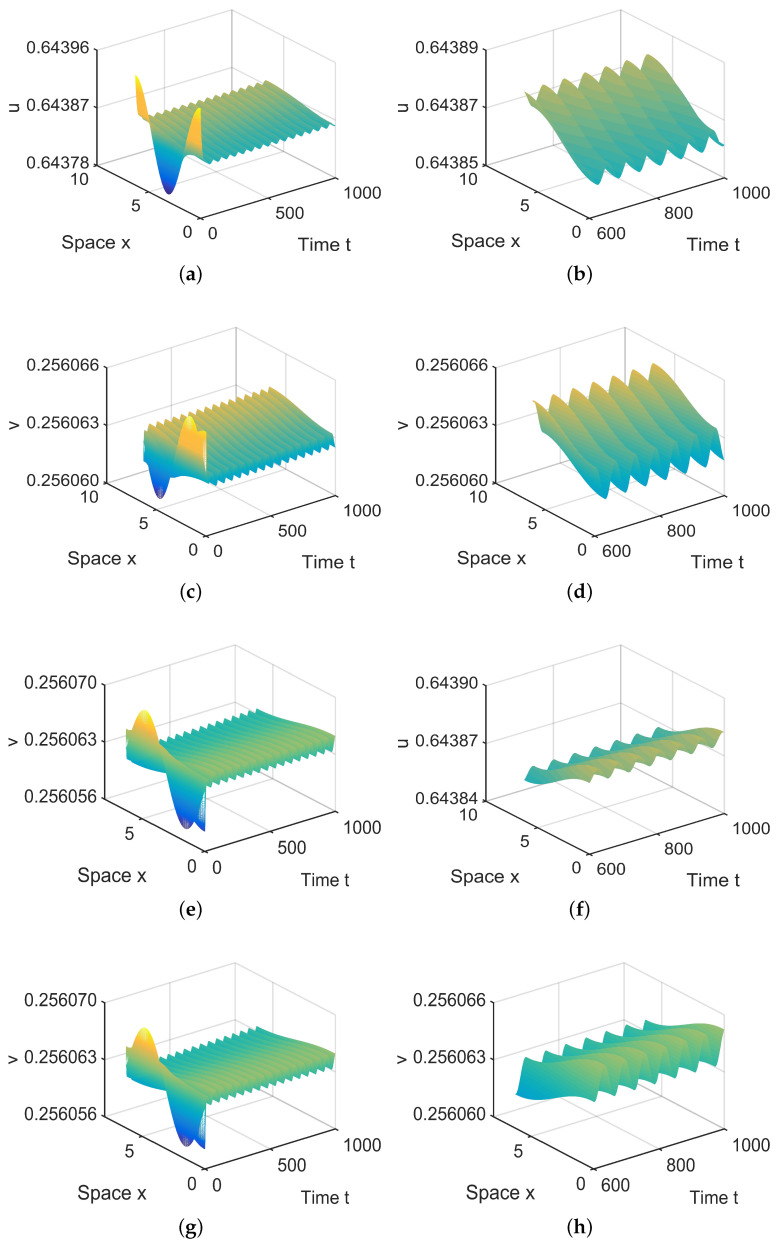
When $\left(\mu_{1},\mu_{2}\right)$ = (−0.000003, 0.0004) lies in $R_{3}$, system (Equation 4) has two spatially inhomogeneous periodic solutions. (**a**,**b**,**e**,**f**) represent the prey *u*, and (**c**,**d**,**g**,**h**) represent the predator *v*. The initial values are $u_{0}\left(x,t\right)=0.6439+0.00008\cos x$ and $v_{0}\left(x,t\right)=0.2561+0.000006\sin x$ in (**a**,**d**) and $u_{0}\left(x,t\right)=0.6439-0.00008\cos x$ and $v_{0}\left(x,t\right)=0.2561-0.000006\sin x$ in (**e**,**f**). (**b**,**d**,**f**,**h**) are long-term behaviors for u and v, respectively.

**Figure 7 entropy-26-00018-f007:**
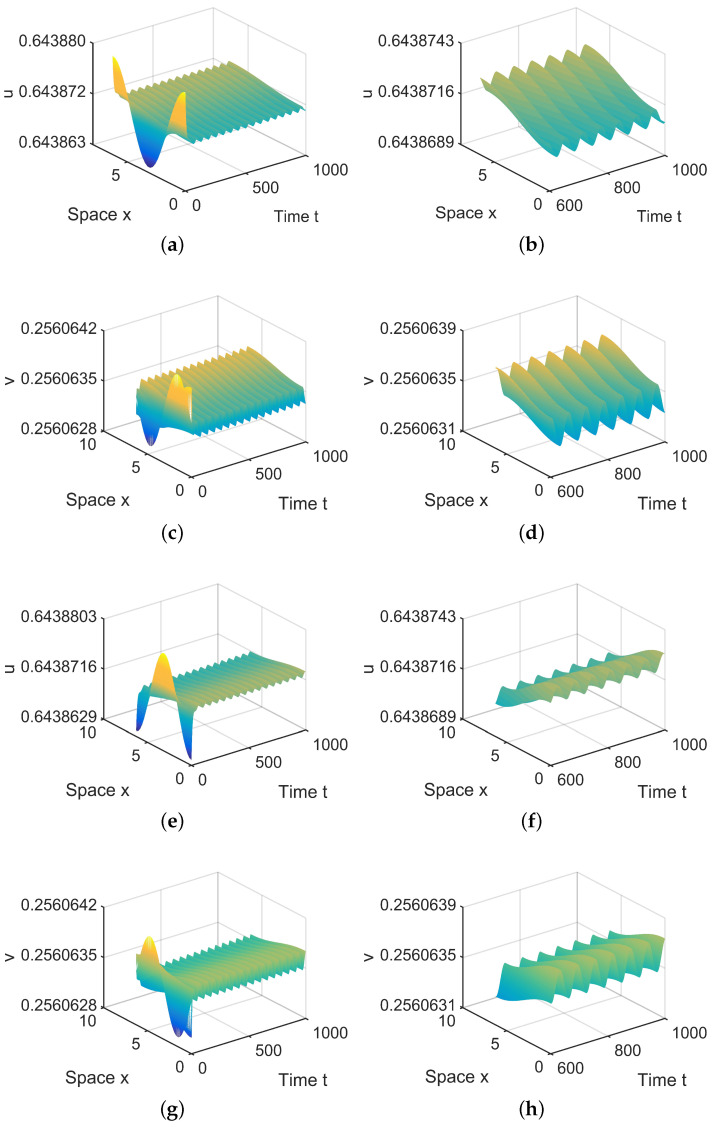
When $\left(\mu_{1},\mu_{2}\right)$ = (−0.000003, 0) lies in $R_{4}$, system (Equation 4) has two spatially inhomogeneous periodic solutions. (**a**,**b**,**e**,**f**) represent the prey *u*, and (**c**,**d**,**g**,**h**) represent the predator *v*. The initial values are $u_{0}\left(x,t\right)=0.6439+0.00008\cos x$ and $v_{0}\left(x,t\right)=0.2561+0.000006\sin x$ in (**a**,**d**) and $u_{0}\left(x,t\right)=0.6439-0.00008\cos x$ and $v_{0}\left(x,t\right)=0.2561-0.000006\sin x$ in (**e**,**f**). (**b**,**d**,**f**,**h**) are long-term behaviors for u and v, respectively.

**Figure 8 entropy-26-00018-f008:**
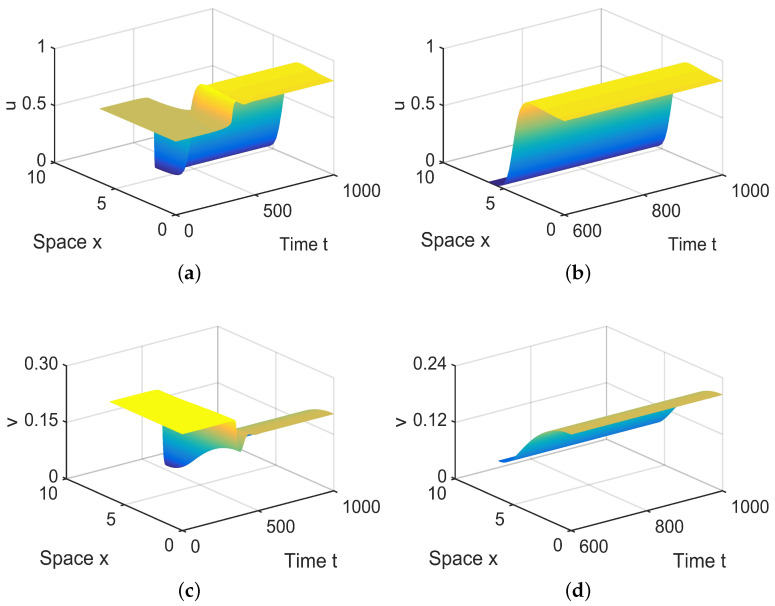
When $\left(\mu_{1},\mu_{2}\right)$ = (−0.000003, −0.1) lies in $R_{5}$, system (Equation 4) has spatially inhomogeneous steady states. (**a**,**b**) represent the prey *u*, and (**c**,**d**) represent the predator *v*. The initial values are chosen as $u_{0}\left(x,t\right)=0.6439-0.00008\cos x$ and $v_{0}\left(x,t\right)=0.2561-0.000006\sin x$ in (**a**–**d**). (**b**,**d**) are long-term behaviors for u and v, respectively.

**Figure 9 entropy-26-00018-f009:**
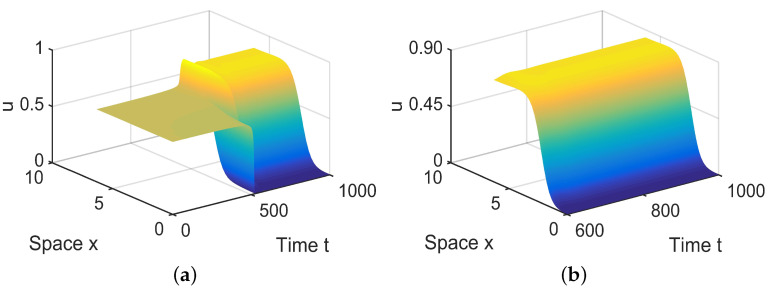
When $\left(\mu_{1},\mu_{2}\right)$ = (−0.0001, 0.04) lies in $R_{6}$, system (Equation 4) has spatially inhomogeneous steady states. (**a**,**b**) represent the prey *u*, and (**c**,**d**) represent the predator *v*. The initial values are chosen as $u_{0}\left(x,t\right)=0.6452-0.0002\sin x$ and $v_{0}\left(x,t\right)=0.2563-0.00025\cos x$. (**b**,**d**) are long-term behaviors for u and v, respectively.

**Table 1 entropy-26-00018-t001:** Stability of the steady states in different regions of system (Equation 28).

Region	Steady States	Stability of the Steady States
$R_{1}$	${\tilde{A}}_{0}$,	${\tilde{A}}_{0}$ is locally asymptotically stable.
$R_{2}$	${\tilde{A}}_{0}$, ${\tilde{A}}_{1}$,	${\tilde{A}}_{1}$ is locally asymptotically stable; ${\tilde{A}}_{0}$ is unstable.
$R_{3}$	${\tilde{A}}_{0}$, ${\tilde{A}}_{1}$, ${\tilde{A}}_{3}^{\pm }$	${\tilde{A}}_{3}^{\pm }$ are locally asymptotically stable; ${\tilde{A}}_{0}$, ${\tilde{A}}_{1}$ are unstable.
$R_{4}$	${\tilde{A}}_{0}$, ${\tilde{A}}_{1}$, ${\tilde{A}}_{2}^{\pm }$, ${\tilde{A}}_{3}^{\pm }$	${\tilde{A}}_{3}^{\pm }$ are locally asymptotically stable; ${\tilde{A}}_{0}$ ${\tilde{A}}_{1}$ and ${\tilde{A}}_{2}^{\pm }$ are unstable.
$R_{5}$	${\tilde{A}}_{0}$, ${\tilde{A}}_{1}$, ${\tilde{A}}_{2}^{\pm }$	${\tilde{A}}_{2}^{\pm }$ are locally asymptotically stable; ${\tilde{A}}_{0}$ ${\tilde{A}}_{1}$ are unstable.
$R_{6}$	${\tilde{A}}_{0}$, ${\tilde{A}}_{2}^{\pm }$	${\tilde{A}}_{2}^{\pm }$ are locally asymptotically stable; ${\tilde{A}}_{0}$ is unstable.

## Data Availability

The data presented in this study are available on request from the corresponding author.

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
