# Peer review of "Turing–Hopf Bifurcation Analysis in a Diffusive Ratio-Dependent Predator–Prey Model with Allee Effect and Predator Harvesting"

_entropy, 2023, doi:10.3390/e26010018_

Round 1

Reviewer 1 Report

Comments and Suggestions for Authors

Manuscript ID: entropy-2760032

Journal: Entropy

Title:  Turing-Hopf bifurcation analysis in a diffusive ratio-dependent predator-prey model with Allee effect and predator harvesting

Authors: Meiyao Chen, Yingting Xu, Jiantao Zhao, Xin Wei

The authors investigate the complex dynamics of a ratio-dependent predator-prey model incorporating Allee effect in prey and predator harvesting. To explore the joint effect of harvesting effort and diffusion on the dynamics of the system, they perform the stability of non-negative constant steady states and obtain sufficient conditions for the occurrence of Hopf bifurcation, Turing bifurcation and Turing-Hopf bifurcation. Numerical simulations are carried out to highlight the theoretical results.

The analysis appears scientifically sound; the mathematical effort is appreciated. Although, in my opinion, the scientific results are not completely innovative, they complement and enrich the results of recent works in the literature on the predator-prey interaction, providing a detailed analysis of Turing-Hopf bifurcation. The analysis presented is well written making the manuscript fluent and readable.

For this reason I recommend a minor revision of the paper.

I suggest a revision of the English throughout the paper.

Some typos need to be corrected and a few minor clarifications need to be made. Just as an example

Line 86: with homogeneous Neumann boundary …;

Line 88: given provided;

Line 153: occurs occur;

Line 168: the expression immediately after line 168 could be eliminated;

and so on.

Comments on the Quality of English Language

I suggest a minor revision of the English throughout the paper.

Author Response

Dear editor, dear reviewers,

We deeply appreciate the time and effort you've spent on reviewing our manuscript. We have studied the valuable comments from you, and tried our best to revise the manuscript. We hope these corrections can meet with your approval. You will find our point-by-point responses to the comments below.

Reviewer 2 Report

Comments and Suggestions for Authors

See attached file.

Comments on the Quality of English Language

Minor editing of English language required.

Author Response

Dear reviewer,

We deeply appreciate the time and effort you've spent on reviewing our manuscript. We have studied the valuable comments from you, and tried our best to revise the manuscript. We hope these corrections can meet with your approval. You will find our point-by-point responses to the comments below.

Reviewer 3 Report

Comments and Suggestions for Authors

see attached pdf

Comments on the Quality of English Language

Generally good. I have shown some minor corrections.

Author Response

(The authors gave the same response as above.)

Reviewer 4 Report

Comments and Suggestions for Authors

The work extends previous research on similar models, providing an analysis of a ratio-dependent predator-prey system with Allee effect, harvesting, and diffusion. The authors explore the dynamics resulting from the interplay of these factors, conducting a thorough theoretical analysis. Stability analyses of interior equilibria, bifurcation analyses, and detailed calculations of Turing-Hopf bifurcation, along with the derivation of the normal form near the singularity, enhance the depth of the analysis. Theoretical results demonstrate that slight changes in harvesting effort and the ratio of diffusion coefficients can destabilize constant steady states, leading to complex spatio-temporal behaviors.

The paper is well-organized and generally well-written, contributing to the understanding of complex ecological systems. However, several minor revisions are warranted for further improvement:

- The numerical simulations should complement the theoretical findings, effectively illustrating the dynamical behaviors predicted by the mathematical model. Unfortunately, the plots are of poor quality, lacking evidence of complex spatiotemporal behaviors. Figures 3-6 suffer from low significance showing just constant dynamics (with also a poor choice of the vertical axis range). The figures need reconsideration and enhancement to better convey research insights.

- Including some details of the numerical methods employed for the simulations would provide additional clarity to the paper.

- The concluding section could be expanded to include suggestions for future research directions, enhancing the completeness of the manuscript.

- To augment the paper's significance, a brief discussion on the practical implications of the study's findings—especially in ecological or biological systems—would be valuable.

In conclusion, the manuscript is quite interesting and well-articulated and  discussed. However, these minor revisions should be addressed before considering its acceptance for publication.

Comments on the Quality of English Language

English is fine and minor editing are required. 

Author Response

Dear reviewer,

We deeply appreciate the time and effort you've spent on reviewing our manuscript. We have studied the valuable comments from you, and tried our best to revise the manuscript. We hope these corrections can meet with your approval. You will find our point-by-point responses to the comments in the PDF below.
